# Regulation of the *Drosophila* ID protein Extra macrochaetae by proneural dimerization partners

Ke Li[1†], Nicholas E Baker[1,2,3]*

[1]Department of Genetics, Albert Einstein College of Medicine, Bronx, United States; [2]Department of Developmental and Molecular Biology, Albert Einstein College of Medicine, Bronx, United States; [3]Department of Ophthalmology and Visual Sciences, Albert Einstein College of Medicine, Bronx, United States

**Abstract** Proneural bHLH proteins are transcriptional regulators of neural fate specification. Extra macrochaetae (Emc) forms inactive heterodimers with both proneural bHLH proteins and their bHLH partners (represented in *Drosophila* by Daughterless). It is generally thought that varying levels of Emc define a prepattern that determines where proneural bHLH genes can be effective. We report that instead it is the bHLH proteins that determine the pattern of Emc levels. Daughterless level sets Emc protein levels in most cells, apparently by stabilizing Emc in heterodimers. Emc is destabilized in proneural regions by local competition for heterodimer formation by proneural bHLH proteins including Atonal or AS-C proteins. Reflecting this post-translational control through protein stability, uniform *emc* transcription is sufficient for almost normal patterns of neurogenesis. Protein stability regulated by exchanges between bHLH protein dimers could be a feature of bHLH-mediated developmental events.
DOI: https://doi.org/10.7554/eLife.33967.001

*For correspondence:
nicholas.baker@einstein.yu.edu

Present address: †Howard Hughes Medical Institute, Department of Physiology, University of California, San Francisco, San Francisco, United States

Competing interests: The authors declare that no competing interests exist.

## Introduction

Proneural bHLH genes play a fundamental role in neurogenesis. Genes from *Drosophila* such as *atonal* (*ato*) and genes of the Achaete-Scute gene complex (AS-C) define the proneural regions that have the potential for neural fate (*Baker and Brown, 2018*; *Bertrand et al., 2002*; *Gómez-Skarmeta et al., 2003*). At least two pathways restrain proneural gene activity and neurogenesis. Lateral inhibition mediated by the Notch pathway blocks neural fate determination by extinguishing proneural gene expression from most proneural cells, with only those that maintain proneural gene expression becoming determined as neural precursors (*Artavanis-Tsakonas et al., 1999*; *Bertrand et al., 2002*). When the Notch pathway is blocked, entire proneural regions can differentiate as neural cells, whereas ectoderm outside of proneural regions is generally unaffected (*Heitzler and Simpson, 1991*). The second restriction is the expression of the Inhibitor of DNA binding (ID) proteins, exemplified in *Drosophila* by Extra macrochaetae (Emc), which contain HLH domains but lack a basic DNA-binding domain (*Benezra et al., 1990*; *Ellis et al., 1990*). Emc (or, in mammals, ID1-4) antagonizes functions of proneural bHLH proteins by forming inactive heterodimers with them (*Benezra et al., 1990*; *Ellis et al., 1990*; *Cabrera et al., 1994*; *Ellis, 1994*; *Norton, 2000*).

The transcription patterns of proneural genes are highly regulated. Surprisingly, therefore, uniform expression of a proneural gene can be sufficient for a normal pattern of neurogenesis (*Rodríguez et al., 1990*; *Brand et al., 1993*; *Domínguez and Campuzano, 1993*; *Usui et al., 2008*). It has been suggested that a pre-exisiting spatial distribution of Emc defines a prepattern of competence for proneural gene function that can help define a restricted pattern of neurogenesis even if

proneural gene transcription is uniform (*Cubas and Modolell, 1992*; *Usui et al., 2008*; *Troost et al., 2015*). Consistent with this model, Emc protein levels are low in proneural regions and neural precursor cells, potentially sensitizing these cells to respond to proneural proteins (*Cubas and Modolell, 1992*; *Bhattacharya and Baker, 2011*; *Troost et al., 2015*). This suggests that the spatial regulation of Emc expression is important for neural patterning.

An important aspect of Emc expression is the regulatory relationship between Emc and Da. Da is required for Emc protein expression, which has been thought to reflect transcriptional regulation (*Bhattacharya and Baker, 2011*). On the other hand, Emc limits Da expression, potentially by heterodimerizing with Da to prevent transcriptional autoregulation of *da* (*Bhattacharya and Baker, 2011*). Variation in Emc levels may be responsible for all spatial distinctions in Da levels, because in the absence of *emc* Da levels are both high and uniform in all *Drosophila* tissues yet examined (*Bhattacharya and Baker, 2011*). Emc can be considered a negative feedback regulator of Da, and some E- and ID- protein genes regulate one another similarly in mammalian cells (*Bhattacharya and Baker, 2011*). Because of their low Emc, most proneural regions express higher levels of Da protein, which is expected to further enhance their competence for productive proneural protein function (*Cronmiller and Cummings, 1993*; *Vaessin et al., 1994*; *Brown et al., 1996*; *Bhattacharya and Baker, 2011*). These reciprocal changes in Emc and Da levels have been seen in proneural cells from all imaginal discs. If Emc expression is restored to proneural regions, changes in Da level do not occur and proneural gene function is affected (*Bhattacharya and Baker, 2011*).

One reason for regulation of Da expression is that proliferating, non-proneural imaginal disc cells cannot tolerate high levels of Da (*Bhattacharya and Baker, 2011*). They respond by regulating the Hippo pathway and other genes to repress cell proliferation and survival (*Andrade-Zapata and Baonza, 2014*; *Wang and Baker, 2015a*). In mammals also, E-proteins and ID-proteins are critical regulators of cell cycle and of cell senescence, and accordingly are tumor suppressors and oncoproteins, respectively (*Perk et al., 2005*; *Lasorella et al., 2014*). In B cells, negative feedback of ID3 on autoregulation of the E-protein TCF3 represents an important barrier to the development of Burkitt's Lymphoma (*Richter et al., 2012*; *Schmitz et al., 2012*). Mammalian E-proteins and ID-proteins are also associated with multiple neurocognitive diseases, including Pitt Hopkins Syndrome, schizophrenia and Rett Syndrome (*Wang and Baker, 2015b*).

Our initial focus was the cross regulation of Emc and Da. The results did not indicate the homeostatic feedback that had been expected, and instead revealed extensive regulation of Emc expression at the level of protein stability, controlled by its binding partners. In most cells, Emc levels were simply matched to Da levels, apparently as a consequence of stabilization of Emc protein by Da. Once proneural genes were expressed, these alternative dimerization partners affected Da and Emc levels in multiple proneural regions including: the morphogenetic furrow of the eye imaginal disc, corresponding to a stripe of cells expressing the proneural gene *ato* that sweeps across the imaginal disc progressively defining the onset of retinal differentiation (*Treisman, 2013*); the wing imaginal disc, where *ac* and *sc* expression specifies dorsal and ventral rows of presumptive sensory bristles of the anterior wing margin(*Skeath and Carroll, 1991*; *Skeath et al., 1994*); the notum region of the wing disc, which differentiates sensory bristles of the adult thorax(*Gómez-Skarmeta et al., 2003*). Consistent with the notion that Emc levels were regulated post-translationally, patterning could continue almost normally when the only source of Emc protein was a uniformly-transcribed transgene. Our findings indicate that Emc (and Da) levels don't define prepattern that precedes regulated proneural gene expression, but are patterned downstream of proneural gene activity. However, proneural proteins are not sufficient to destabilize Emc proteins at all locations. Therefore, other mechanisms must exist that contribute to define the proneural prepattern.

## Results

### Da and emc protein levels are proportional to da gene dose

Da and Emc show fairly uniform protein levels in most imaginal disc cells but they change dynamically in proneural regions (*Cronmiller and Cummings, 1993*; *Brown et al., 1995*; *Bhattacharya and Baker, 2011*)(*Figure 1A–B*). We first investigated the non-proneural regions. If Da and Emc levels were kept even by homeostatic negative feedback, they should compensate for modest changes in expression levels. We decided to compare protein levels in cells homozygous for a null allele

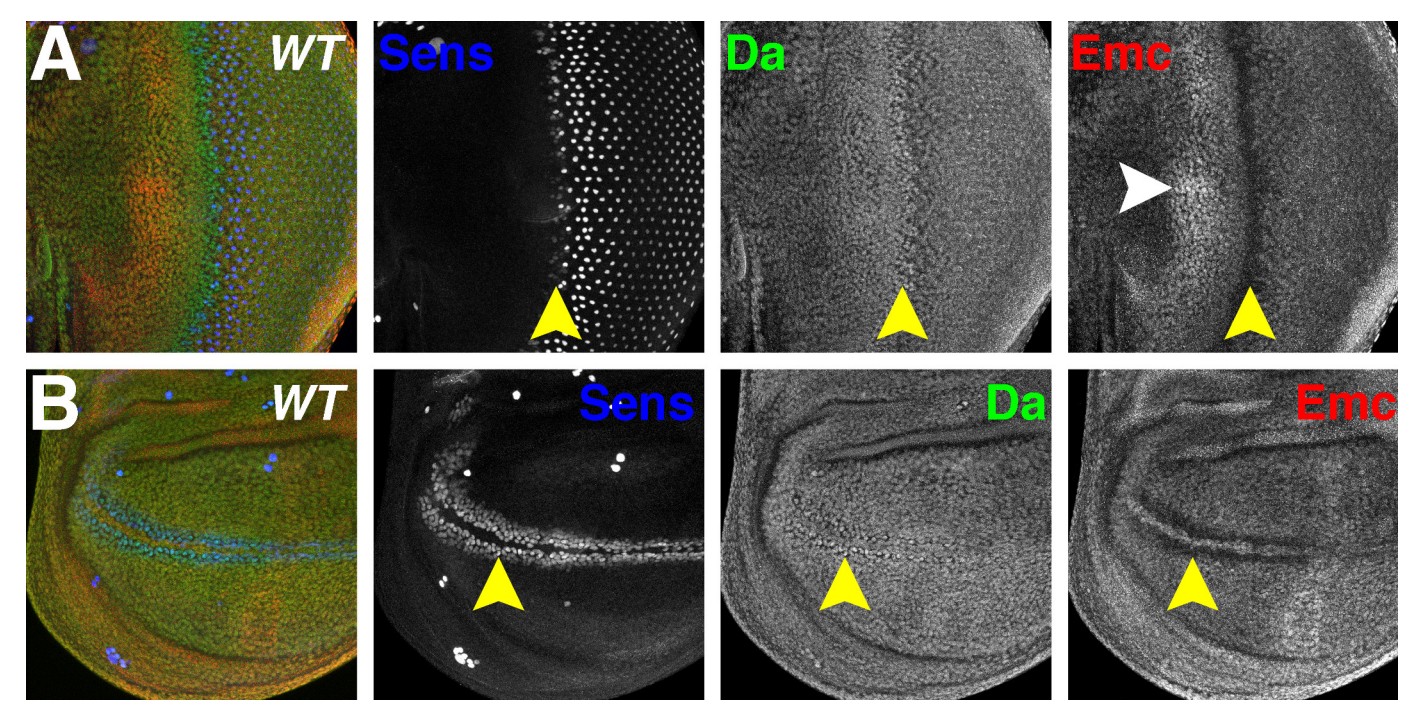

**Figure 1.** Da and Emc are broadly expressed proteins that are modulated in proneural regions. Panels show immunofluorescence labeling of *Drosophila* eye imaginal discs (**A**) and wing imaginal discs (**B**). Yellow arrowheads indicate the morphogenetic furrow of the eye disc (**A**) and future anterior wing margin of the wing disc (**B**). Neural precursor cells arise in those proneural regions and are labeled by Sens (blue). Da (green) and Emc (red) proteins are broadly detected. Da levels are elevated within proneural regions whereas Emc levels are reduced. At the wing margin Da may be elevated in fewer cells than those where Emc is reduced. In addition, higher Emc protein levels are often higher in the equatorial region of the anterior eye disc (white arrowhead, panel A;see text). Genotype: $w^{1118}$.

DOI: https://doi.org/10.7554/eLife.33967.002

('clones'), heterozygous for a null allele and a wild type allele (non-recombined cells), and homozygous for wild type alleles ('twin-spots'). Since the *da* and *emc* genes map to different chromosomes, parallel mitotic recombination of both chromosomes can generate up to nine different genotypes in the same tissue (1. $da^{-/-}$; $emc^{-/-}$, 2. $da^{-/-}$; $emc^{-/+}$, 3. $da^{-/-}$; $emc^{+/+}$; 4. $da^{-/+}$; $emc^{-/-}$, 5. $da^{-/+}$; $emc^{-/+}$, 6. $da^{-/+}$ $emc^{+/+}$, 7. $da^{+/+}$; $emc^{-/-}$, 8. $da^{+/+}$; $emc^{-/+}$, 9. $da^{+/+}$; $emc^{+/+}$). Comparing Da and Emc expression in those genotypes in parallel should reveal any homeostatic regulation.

Unless otherwise regulated, gene expression is proportional to gene copy number (*Ciferri et al., 1969*). This was the case for GFP expressed from the *[Ubi-GFP]* transgene. Mitotic recombination in *[Ubi-GFP]* transgene heterozygotes led to clones with 0 or two transgene copies in the background of cells with one copy. GFP fluorescence intensity from confocal images was proportional to *[Ubi-GFP]* copy number (*Figure 2A–B*). When GFP expression was instead detected using indirect immunohistochemistry with an anti-GFP antibody, this signal was also proportional to *[Ubi-GFP]* copy number and to GFP fluorescence (*Figure 2A–B*). Thus, immunostaining and confocal microscopy were consistent with linear detection of protein expression levels in wing imaginal discs.

Next, mitotic recombination was induced in the $da^{-/+}$; $emc^{-/+}$ genotype and Da protein levels were quantified in cell populations with different doses of the wild type *da* and *emc* genes. Contrary to the idea that uniform levels of Da protein were subject to homeostatic feedback, Da protein levels were instead proportional to *da* gene dose (*Figure 2C,E* and *Figure 2—figure supplement 1A,C*). In the background wild type for *emc* (i.e. $emc^{+/+}$), cells with two copies of the wild type *da* gene had almost twice as much Da protein as cells with only one copy (*Figure 2E* and *Figure 2—figure supplement 1A,C*). This did not support the notion that the level of Da expression was buffered by negative feedback regulation from *emc*. Accordingly, Da protein levels did not change when one copy of *emc* was removed, that is Da protein levels were indistinguishable in the $emc^{+/+}$ and $emc^{-/+}$

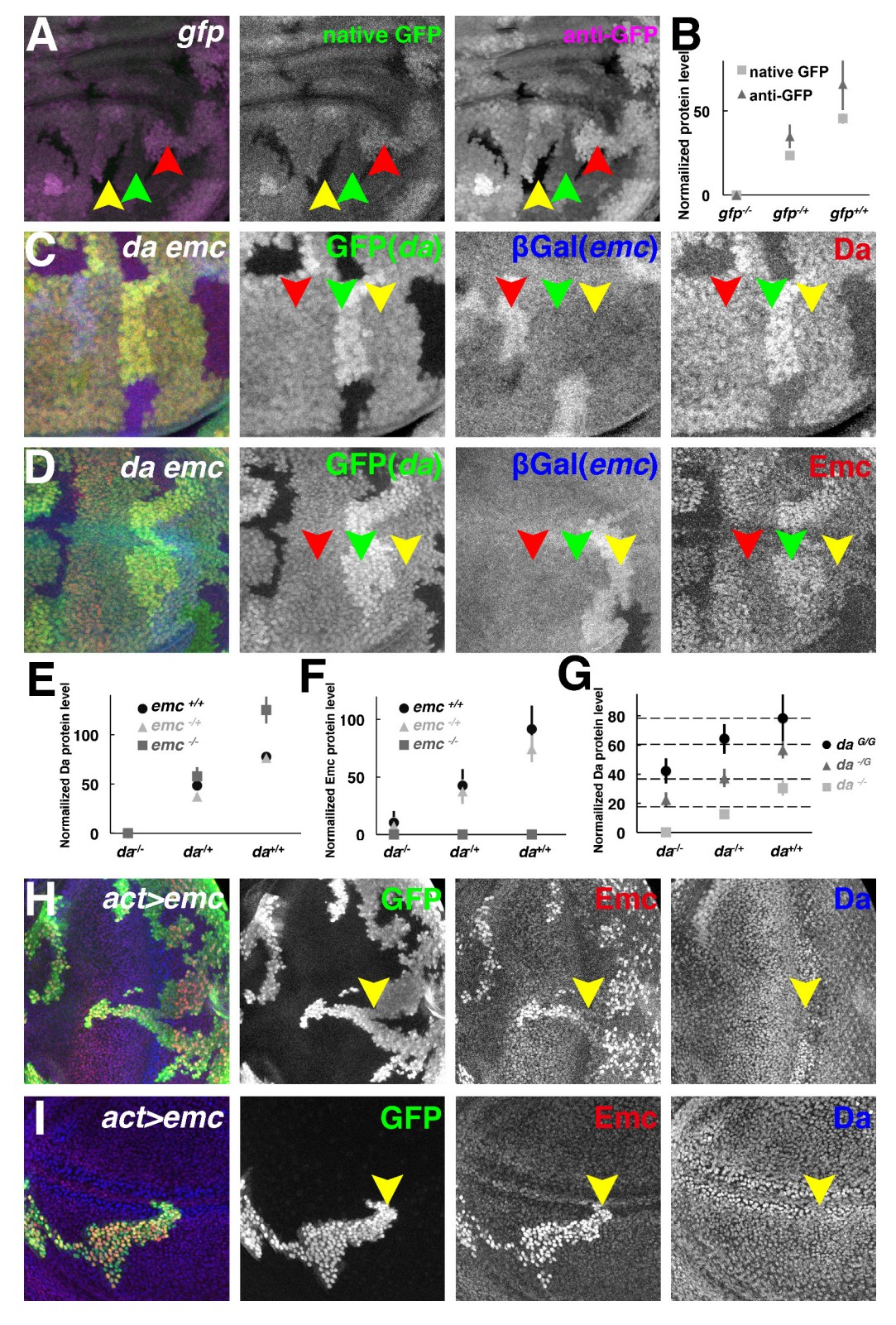

**Figure 2.** Both Da and Emc protein levels depend on *da* gene dose. (A) GFP signals in wing imaginal disc mosaic for the *ubi-GFP* transgene detected simultaneously by native GFP fluorescence (green) and by anti-GFP antibody (magenta). (B) Quantification of native GFP signal and anti-GFP antibody signal in (A), showing their linearity to the *gfp* gene dose (N = 4). Panels (C–D) show mosaic imaginal disc tissues obtained after mitotic recombination of heterozygous genotypes (see text). Homozygous *da* and *emc* mutant clones are negatively marked by GFP (green) or βGal (blue), respectively, within

*Figure 2 continued on next page*

*Figure 2 continued*

the same wing disc. Reciprocal twin spots are brightly labeled while unrecombined heterozygous cells show intermediate labeling. (C) Anti-Da labeling (red) in cells with different *da* and *emc* gene copies. Note that $da^{-/+}$ $emc^{+/+}$ (red arrowheads) and $da^{-/+}$; $emc^{-/+}$ (yellow arrowheads) cells have indistinguishable levels of Da protein. Cells with higher *da* gene dose have more Da protein (green arrowheads). (D) Anti-Emc labeling (red) in cells with different *da* and *emc* gene copies. Note that $da^{-/+}$; $emc^{-/+}$ (red arrowheads) and $da^{-/+}$ $emc^{+/+}$ (yellow arrowheads) have similar Emc protein levels, while $da^{+/+}$; $emc^{-/+}$ (green arrowheads) cells express higher levels of Emc than $da^{-/+}$; $emc^{-/+}$ (red arrowheads) cells do. (E–F) Quantification of Da (E) and Emc (F) antibodies fluorescence intensities. Mean ± SEM is shown (N = 7). X-axes represent the endogenous *da* gene dose and different colors represent different *emc* gene dose (E–F). In panel (E), the Da level in $da^{+/-}$; $emc^{+/+}$ cells appears greater than half that in to $da^{+/+}$; $emc^{+/+}$ cells but this was not reproduced in other studies (see panel G and *Figure 2—figure supplement 1C*). The Da level in $da^{+/-}$; $emc^{+/+}$ cells was not statistically different from that in $da^{+/-}$; $emc^{+/-}$ cells. Remarkably, $da^{+/+}$; $emc^{-/+}$ cells have higher Emc protein levels than $da^{-/+}$ $emc^{+/+}$ cells do (p=0.00068, two-tailed t-test). (G) Quantification of Da antibody labeling from mosaic wing discs where clones vary the copy number of the endogenous *da* locus from 0 to 2 and independently vary the copy number of an unlinked genomic rescue transgene from 0 to 2 (see text and *Figure 2—figure supplement 1E*). X-axis represents the endogenous *da* gene dose and different colors represent *da* rescue transgene dose. Dashed horizontal lines represent mean Da levels for 1,2,3 and 4 gene copy genotypes. Note that the genomic transgene consistently expresses more Da protein than the endogenous locus. Mean ± SEM is shown,(N = 10). (H–I) Random flip-on clones overexpressing *emc* using a *UAS-emc* line are marked by GFP (green). Emc (red) over-expression abolishes Da (blue) upregulation in the morphogenetic furrow of the eye disc (H, yellow arrowheads) and the presumptive wing margin in the wing disc (I, yellow arrowheads), but has no discernible effect elsewhere. Genotypes: (A) *hsFLP; Ubi-GFP FRT40/FRT40*, (C–D) *hsFLP; da³ FRT40/Ubi-GFP FRT40; emc^AP6 FRT80/arm-LacZ FRT80*, (H–I) *hsFLP; UAS-emc 5.3/+; act > CD2>Gal4, UAS-GFP/+*.

DOI: https://doi.org/10.7554/eLife.33967.003

The following figure supplement is available for figure 2:

**Figure supplement 1.** Panels A and B show gene dose mosaic experiments like those of *Figure 2* except that the dose of only *da* (A) or *emc* (B) is varied.

DOI: https://doi.org/10.7554/eLife.33967.004

backgrounds, so long as *da* gene copy number was the same (*Figure 2C,E* and *Figure 2—figure supplement 1B,D*). In the total absence of *emc* (i.e. $emc^{-/-}$), Da levels were elevated, as reported previously (*Bhattacharya and Baker, 2011*)(*Figure 2E* and *Figure 2—figure supplement 1B,D*). We extended these observations using a genomic rescue transgene to vary *da* copy number from 0 to 4. Extra *da* gene dose increased Da protein levels linearly (*Figure 2G* and *Figure 2—figure supplement 1E*). In summary, Da expression was proportional to *da* gene dose and unaffected by *emc* gene dose unless the *emc* gene was completely deleted. This suggested that Da autoregulation was not significant at the Emc levels normal for imaginal disc cells outside proneural regions.

We used Emc over-expression to look for Da regulation in another way. Outside proneural regions, Da levels were not affected by Gal4-driven Emc (*Figure 2H–I*). Ectopic Emc only reduced Da levels in the proneural cells of the morphogenetic furrow and wing margin, where endogenous Emc is very low (*Bhattacharya and Baker, 2011*)(*Figure 2H–I*). Since Emc is expected to prevent autoregulatory *da* expression, this result suggests three things: that *da* does not autoregulate outside of proneural regions; that Emc is already in excess outside of proneural regions; and that feedback from Emc is not what maintains steady Da levels.

Emc protein levels were measured in the same *da* and *emc* genetic combinations resulting from mitotic recombination in the $da^{-/+}$; $emc^{-/+}$ genotype (*Figure 2D,F*). Remarkably, Emc levels were also proportional to *da* gene dose, and cells with the same *da* gene dose (and therefore the same Da protein level) had indistinguishable Emc protein levels, regardless of whether one or two copies of the *emc* genes was present (*Figure 2F* and *Figure 2—figure supplement 1B,D*). These observations were exemplified by the finding that $da^{+/+}$; $emc^{-/+}$ cells had more Emc protein than $da^{-/+}$; $emc^{+/+}$ cells, despite the double *emc* gene dose in the latter (*Figure 2D,F*). As reported previously, $da^{-/-}$ cells expressed only low levels of Emc protein (*Bhattacharya and Baker, 2011*)(*Figure 2F* and *Figure 2—figure supplement 1A,C*). In summary, Emc protein levels did not seem to be buffered against fluctuations in Da levels, in fact *da* gene dose, rather than *emc* gene dose, was the determinant of Emc protein level.

We focused on the wing disc for quantification of Emc and Da levels since it mostly consists of similar cells, developing synchronously (see Materials and methods for details). Similar results were observed in eye discs, although we did not perform quantitative analysis because of the multiple cell types and developmental stages present in eye discs.

## Emc is stabilized by Da in S2 cells

Emc dimerizes with bHLH proteins, including Da, through HLH-mediated interactions (*Van Doren et al., 1991*; *Cabrera et al., 1994*). Our observations on Emc levels could be explained if Emc protein was unstable except in a heterodimer with Da. To test this, the half-life of Emc was measured in cultured S2 cells.

A V5-tagged Emc open reading frame cloned under the control of an actin promoter was transiently transfected into S2 cells. The expression of the full-length Emc protein was confirmed by western blot analyses. Emc protein half-life was estimated by following a time course after cycloheximide (CHX) addition to block new protein synthesis. Emc was a short-lived protein with half-life around 30 min (*Figure 3A,E*). Treatment of the cells with the proteasome inhibitor MG132 significantly extended the half-life of Emc to more than 300 min (*Figure 3B,E*). Therefore, in S2 cells Emc was an unstable protein degraded via the proteasome-dependent pathway.

Cotransfection of Flag-tagged Da rendered Emc very stable (*Figure 3D–E*), increasing the half-life of Emc at least as much as blocking proteasomal degradation (*Figure 3E*). Similar studies showed that Da itself was a stable protein (*Figure 3C,F*) although Da stability might be somewhat shortened by costransfection with Emc (*Figure 3D,F*). The half-life of Da alone was estimated at > 300 min, but that of Da co-transfected with Emc at 139 min (*Kiparaki et al., 2015*)(*Figure 3D, F*). It would be interesting to investigate whether this difference affects Da levels in vivo when *emc* is mutated.

To verify these findings in vivo, Da was overexpressed in wing imaginal discs using the Gal4 system. Da overexpression led to Emc protein accumulation in exactly the same cells (*Figure 3G*).

Altogether, these data suggested Emc becomes stabilized in Da/Emc heterodimers. This could explain both why Emc protein levels depend on Da levels and are relatively homogenous outside proneural regions, rather than transcriptional regulation of *emc* by *da*, as suggested previously (*Bhattacharya and Baker, 2011*). Emc might also affect Da stability, to a lesser degree.

## Ato is required for altering Da and Emc levels in the morphogenetic furrow

Emc instability could also explain its reduction in proneural regions. Da might become limiting where Da also heterodimerizes with proneural proteins. Significantly, Dpp and Hh, the same signals that induce Ato expression in the morphogenetic furrow, are also required to change Da and Emc levels (*Greenwood and Struhl, 1999*; *Curtiss and Mlodzik, 2000*; *Bhattacharya and Baker, 2011*), consistent with the possibility that destabilization of Emc is linked to Ato expression.

Apparently contradicting this idea, however, Da and Emc levels continue to change in the morphogenetic furrow in clones of cells homozygous for the *ato¹* mutation (*Bhattacharya and Baker, 2011*). The *ato¹* mutation contains three coding substitutions, A25T, K253N and N261I (*Jarman et al., 1994*)(*Figure 4A*). K253 and N261 lie in the basic domain that is required for DNA-binding (*Figure 4A*). The *ato¹* allele has been considered genetically amorphic, since its effects on neurogenesis resemble that of a deletion of the gene (*Jarman et al., 1994*), but it still encodes detectable protein that is expected to contain a helix-loop-helix domain and therefore may be able to heterodimerize with Da (*Figure 4—figure supplement 1A*) (*Jarman et al., 1995*). To characterize a true protein null allele we determined the sequence of *ato³*, which has the same loss-of-function phenotype as *ato¹* with respect to neurogenesis but does not encode detectable protein (*Jarman et al., 1995*)(*Figure 4—figure supplement 1B*). Sequencing of *ato³* genomic DNA revealed a single base-pair change 8278687C > T that introduced a premature stop codon (Q188X) upstream of the bHLH domain (*Figure 4A*). Therefore even if the *ato³* mutant cells contain a protein not detected by the available antibody, this protein should lack the bHLH domain and thus not be able to form heterodimers.

As reported previously, cells homozygous for *ato¹* mutant downregulated Emc (*Figure 4B*) and upregulated Da (*Figure 4C*) in the morphogenetic furrow, like wild type cells (*Bhattacharya and Baker, 2011*)(*Figure 1A*). By contrast, cells homozygous for *ato³* retained Emc in the morphogenetic furrow (*Figure 4D*) and failed to upregulate Da (*Figure 4E*). Thus, *ato* function does regulate Da and Emc expression levels in the morphogenetic furrow, but independently of aspects of *ato* function altered in the *ato¹* allele, which behaves as a null allele for neurogenesis. We have been unable to

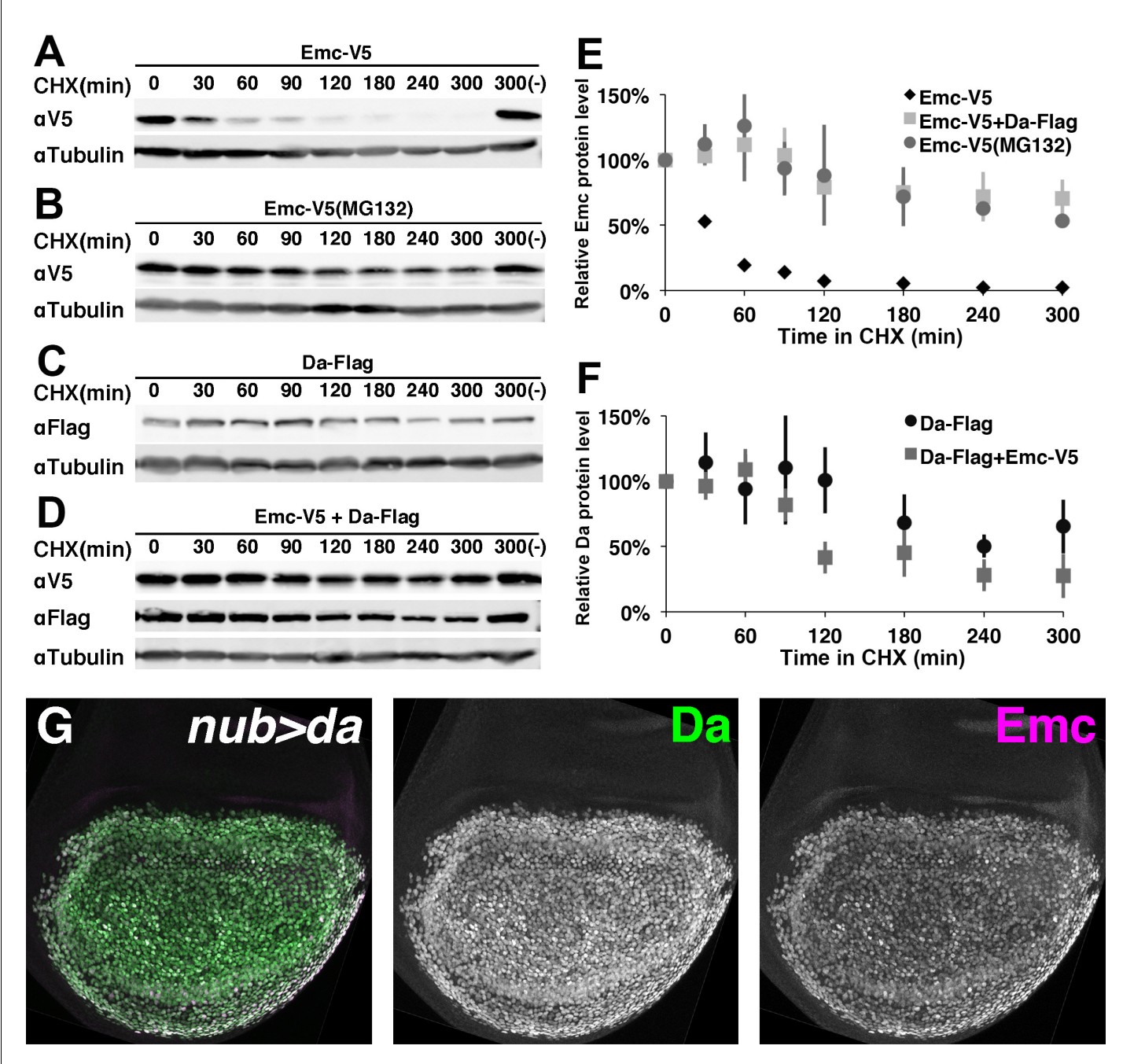

**Figure 3.** Emc is unstable alone but stabilized in the presence of Da. In panels (**A-F**) proteins from S2 cells were analyzed by western blot following a time course after cycloheximide (CHX) addition. (-) indicates absence of CHX treatment. αTubulin is used as a loading control. (**A**) Western blot of Emc-V5 show Emc had a short life in S2 cells. (**B**) Cells are pretreated with a proteasomal inhibitor MG132 to block ubiquitin-proteasome mediated degradation before CHX addition. Emc-V5 degradation is significantly slower. (**C**) Western blot of Da-Flag show Da is a very stable protein in S2 cells. (**D**) Co-transfection of Da-Flag with Emc-V5 make Emc a stable protein, while the half-life of Da seems shorter in the presence of Emc. (**E**) Quantification of Emc-V5 half-lives in panels A,B and D. Mean ± SEM is shown and calculated from 3 to 5 biological replicates(ie independent transfections). All experiments performed were included for quantification. (**F**) Quantification of Da-Flag half-lives in panels C and D. Mean ± SEM is shown and calculated from 3 to 5 biological replicates. All experiments performed were included for quantification. (**G**) Expression of Da in the developing wing disc using nub-Gal4 drives a high level of Da protein (green: image underexposes the normal Da expression in surrounding cells in order to record the Da over-expression). Emc protein (magenta) is stabilized in a precisely corresponding pattern (normal Emc expression in surrounding cells underexposed). Note that Da over-expression is likely also to increase transcription of the endogenous *emc* gene(*Bhattacharya and Baker, 2011*).
DOI: https://doi.org/10.7554/eLife.33967.005

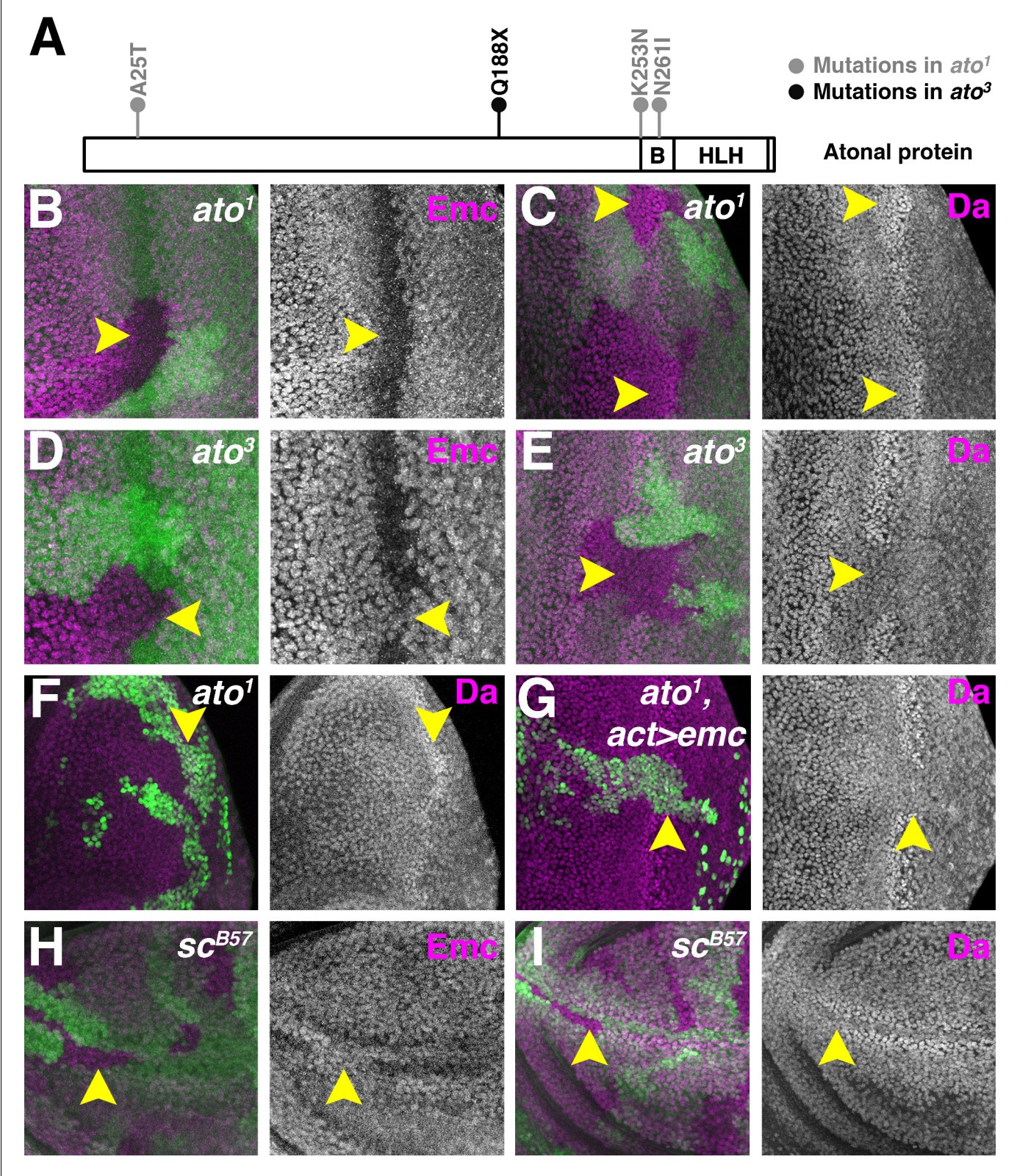

**Figure 4.** Proneural proteins are necessary for modulating Da and Emc levels. (**A**) Cartoon of Ato protein showing sequence changes in *ato¹* and *ato³* mutants. 'B' indicating the basic domain and 'HLH' indicating the helix-loop-helix domain of wild type Ato. (**B–E**) Homozygous *ato¹* (**B and C**) or *ato³* (**D and E**) mutant clones are marked by the absence of βGal (green). (**B**) Emc (magenta) goes down in *ato¹* clones in the furrow (arrow). (**C**) Da (magenta) goes up in *ato¹* clones in the furrow (arrow), at levels comparable to (if not higher) than the normal high level of Da in the furrow. (**D**) Emc (magenta) is
*Figure 4 continued on next page*

Figure 4 continued

retained in *ato³* clones in the furrow (arrow), at levels comparable to the normal Emc levels ahead of the furrow. (E) Da (magenta) fail to upregulate in *ato³* clones in the furrow (arrow). (F–G) *ato¹* MARCM clones are positively labeled by GFP. (F) cells homozygous for *ato¹* mutant (arrow) upregulatd Da (magenta). (G) Overexpression of Emc in *ato¹* mutant clones prevents Da upregulation (arrow). (H–I) Homozygous *sc^B57* mutant clones are marked by the absence of GFP (green). (H) Emc (magenta) is retained in cells lacking all the four AS-C genes in the wing margin of the wing discs (arrow). (I) Da (magenta) is not elevated in wing margin cells homozygous for the AS-C mutant (arrow). Genotypes: (B–C) *hsFLP; FRT82 ato¹/FRT82 arm-lacZ*; (D–E) *hsFLP; FRT82 ato³/FRT82 arm-lacZ*; (F) *hsFLP, UAS-GFP; tub-Gal4/+; FRT82 tub-Gal80/FRT82 ato¹*; (G) *hsFLP, UAS-GFP; tub-Gal4/UAS-emc 5.3; FRT82 tub-Gal80/FRT82 ato¹*; (H–I) *Df(1)sc^B57FRT101/Ubi GFP FRT101; hsFLP/+.*.

DOI: https://doi.org/10.7554/eLife.33967.006

The following figure supplement is available for figure 4:

Figure supplement 1.

DOI: https://doi.org/10.7554/eLife.33967.007

express Ato in S2 cells and therefore could not test whether Ato destabilizes Emc directly or by sequestering Da.

To gain further insight into the upregulation of Da that occurs in *ato¹* clones, we performed MARCM (*Lee and Luo, 1999*; *Lee and Luo, 2001*) experiments to overexpress Emc in *ato¹* mutant cells. Unlike plain *ato¹* mutant cells (*Figure 4C,F*), cells homozygous for *ato¹* and also overexpressing Emc failed to upregulate Da and maintained pre-existing Da levels (*Figure 4G*). This suggests that Da upregulation is bHLH-mediated, for example by transcriptional autoregulation of the *da* gene, or by greater stability of Da-Da and Da-Ato1 dimers in the absence of Emc.

## AS-C is required for altering Da and Emc levels in wing disc proneural regions

Like the morphogenetic furrow, proneural cells of the anterior wing margin also elevate Da and reduce Emc (*Bhattacharya and Baker, 2011*) (*Figure 1B*). When all the four AS-C bHLH genes were deleted, Da was no longer elevated at the wing margin (*Figure 4I*) and Emc was not downregulated (*Figure 4H*). Thus AS-C gene function regulates Da and Emc levels in the anterior wing margin, as *ato* does in the eye disc.

It has been reported that Emc regulation is independent of AS-C in the notum primordium of the wing disc (*Cubas and Modolell, 1992*; *Troost et al., 2015*). These studies used the viable mutation *sc^10-1*, which deletes *ac* but not *l'sc* or *ase*, and truncates the C-terminus of Sc after the penultimate residue of the bHLH domain (*Villares and Cabrera, 1987*; *Rodríguez et al., 1990*). Although *sc^10-1* behaves genetically as a mutation of both *ac* and *sc*, it has the potential to encode a truncated Sc protein that includes much of the HLH domain (*Villares and Cabrera, 1987*). At the anterior wing margin, *sc^10-1* appeared to present an intermediate phenotype between wild type (*Figure 1B*) and *sc^B57* (*Figure 4H–I*), expressing Da but to a lower degree than wild type, and retaining more Emc expression than wild type (*Figure 5A*), although a clonal analysis would be useful to confirm this impression.

The notum differentiates a number of innervated bristles derived from individual sensory organ precursor (SOP) cells which express elevated Da and low Emc (*Bhattacharya and Baker, 2011*). In addition, other more subtle differences in Emc occur (*Figure 5B*)(*Troost et al., 2015*). Emc expression is higher in a large domain along the anterior margin, and two small ventral domains located posteriorly and centrally (*Figure 5B*). Proneural regions, which can be identified by Sca-LacZ (*Mlodzik et al., 1990*), lie in between these higher Emc domains (*Troost et al., 2015*). These Emc domains are not affected by *sc^10-1* (*Troost et al., 2015*) (*Figure 5D*) or by *sc^B57* clones (*Figure 5C*). By contrast, we did not succeed in locating cells lacking Emc at the locations of the missing SOP cells (*Figure 5C–D*). Although it might be difficult to locate individual cells lacking Emc expression in the absence of any SOP marker, we also did not see cells with higher Da, which would be expected if regulation of Emc and Da was independent of AS-C (*Figure 5C–D*), suggesting the AS-C may regulate Emc and Da levels in the precursors of thoracic macrochaetae as well as at the anterior wing margin.

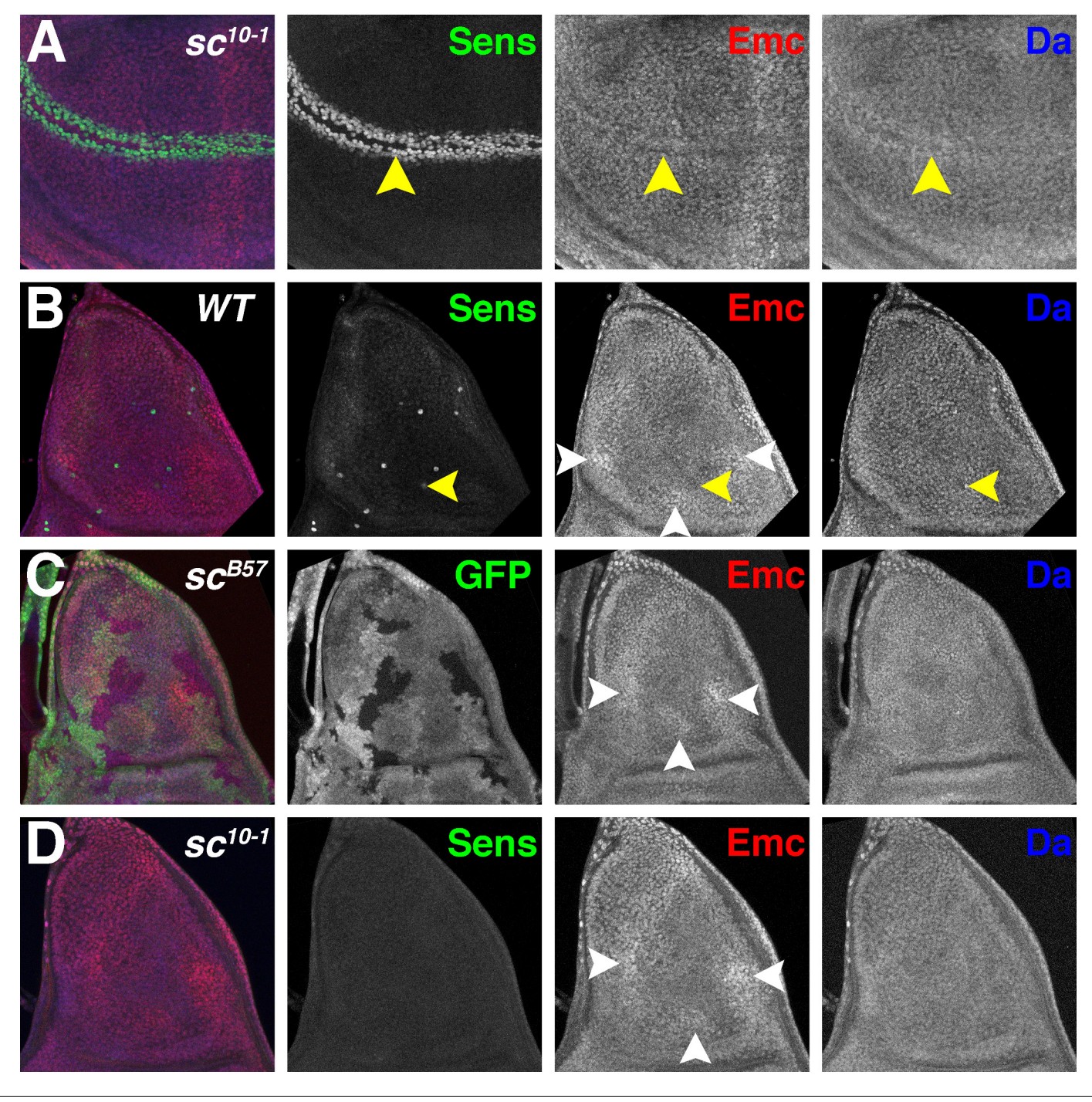

**Figure 5.** Proneural genes regulate Emc and Da in the notum. (A) In $sc^{10-1}$ wing discs, fewer cells are labeled by Sens (green) in the wing margin (yellow arrowheads). Emc (red) and Da (blue) levels are not affected as strongly as in the wild type (compare *Figure 1B*). (B) In wild type notum, Sens (green) marks single sensory organ precursor (SOP) cells. Emc (red) and Da (blue) proteins are expressed widely, although the SOP cells generally have lower Emc and higher Da (eg yellow arrowheads)(*Bhattacharya and Baker, 2011*). In addition, Emc protein levels are higher in particular domains (white arrowheads). High-Emc domains lack SOP cells. (C) Homozygous $sc^{B57}$ clones lack GFP. SOP cells with lower Emc(red) and higher Da(blue) were never observed in $sc^{B57}$ clones, although the regions of high Emc were unchanged (white arrowheads). (D) The whole $sc^{10-1}$ notum lacks Sens positive SOP cells (green). SOP cells with lower Emc(red) and higher Da(blue) were never observed in $sc^{10-1}$notum, although the regions of high Emc were unchanged (white arrowheads). (A, D) $Df(1)sc^{10-1}$; (B) $w^{1118}$; (C) $Df(1)sc^{B57}FRT101/Ubi GFP FRT101; hsFLP/+$.

DOI: https://doi.org/10.7554/eLife.33967.008

## Proneural genes regulate Emc levels post-transcriptionally

To further investigate how proneural proteins regulate Emc expression in proneural regions, the regulation of *emc* transcription was examined. Three enhancer trap lines, *emc-GFP^YB0040^*, *emc-GFP^YB0067^* and *emc^04322^* largely recapitulating the mRNA distribution (*Figure 6A* and data not shown)(*Baonza et al., 2000*; *Baonza and Freeman, 2001*; *Bhattacharya and Baker, 2009*; *Spratford and Kumar, 2015*). They exhibited reduced expression in the morphogenetic furrow and the anterior wing margin (*Figure 6A*).

In both *ato^1^* and *ato^3^* mutant clones, *emc* reporter expression remained low in the morphogenetic furrow region (*Figure 6B* and *Figure 6—figure supplement 1*). Mutant cells posterior to the furrow also exhibited lower reporter expression (*Figure 6B* and *Figure 6—figure supplement 1*), possibly due to eye differentiation being prevented by *ato* mutations (*Jarman et al., 1994*). These data indicated *ato* was not required to repress *emc* transcription in the morphogenetic furrow. Therefore, regulation of Emc expression by Ato was post-transcriptional, like regulation of Emc expression by Da.

## Proneural genes are not sufficient to regulate Da or Emc protein levels

If proneural proteins destabilize Emc by sequestering Da, then ectopic expression of Ato (or AS-C proteins) should have this effect in other, non-proneural regions of imaginal discs. Gal4-mediated Ato expression was driven in clones of cells to test this. When HA-tagged Ato was induced in eye discs or wing discs clones, Da was slightly upregulated but Emc was not reduced (*Figure 7A* and *Figure 7—figure supplement 1A*). Similar results were obtained with weaker expression of untagged Ato from a different transgene (*Figure 7—figure supplement 1B*). The levels of ectopic HA-tagged Ato were generally similar to the endogenous levels in the morphogenetic furrow of wild type eye discs (*Figure 7B*) and in many cases were sufficient to express *scabrous*, a general reporter of proneural gene activity (*Mlodzik et al., 1990*)(*Figure 7E*). The ectopic Ato levels were somewhat heterogenous, however, with individual clones containing cells with higher and lower Atonal in a salt-and-pepper fashion. We measured the levels of Emc and Da in individual cells with different Ato levels of ectopic Ato, without observing any correlation (*Figure 7C–D* and *Figure 7—figure supplement 1C–D*). Interestingly, the GAL4-induced expression level of ectopic Ato was lower in a region spanning the morphogenetic furrow (*Figure 7B*).

Consistent with previous conclusions (*Bhattacharya and Baker, 2011*), Ato expression by itself was insufficient to induce premature neuronal differentiation anterior to the morphogenetic furrow

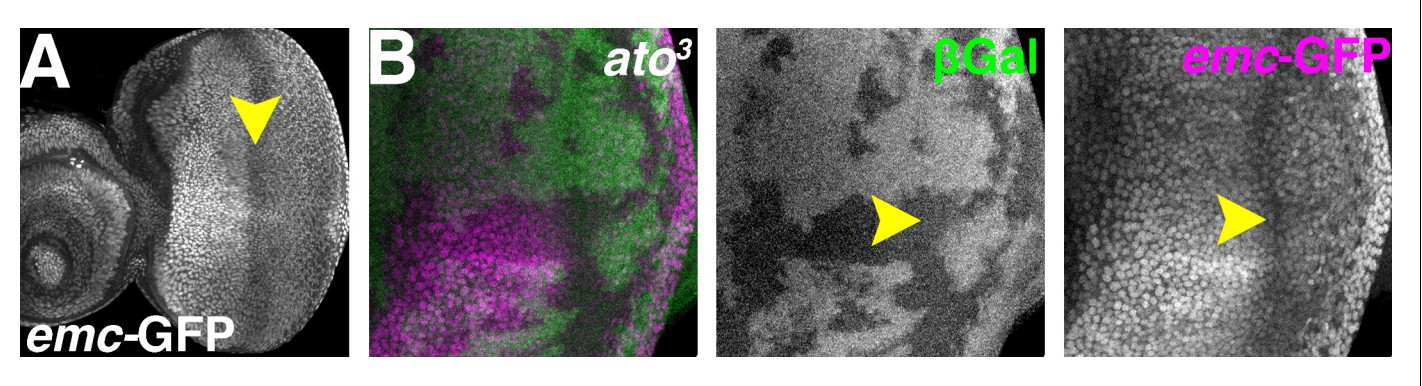

**Figure 6.** Atonal regulates Emc expression post-transcriptionally. (**A**) *emc* enhancer trap expression in the eye disc in the *emc-GFP^YB0067^* line. Downregulation in the morphogenetic furrow (arrow) is broader and less complete than seen fpr the Emc protein (compare *Figure 1A*). (**B**) *ato^3^* mutant clones are marked by the absence of βGal (green). *emc* enhancer trap (magenta) is lower in the furrow both inside and outside *ato^3^* mutant clones. Genotypes: (**A**) *emc-GFP^YB0067^*; (**B**) *eyFLP; emc-GFP^YB0067^, FRT82 arm-lacZ/FRT82 ato^3^*.
DOI: https://doi.org/10.7554/eLife.33967.009

The following figure supplement is available for figure 6:

**Figure supplement 1.** *ato^1^* mutant clones are marked by the absence of βGal (green) and *emc* enhancer trap is detected by GFP (magenta).
DOI: https://doi.org/10.7554/eLife.33967.010

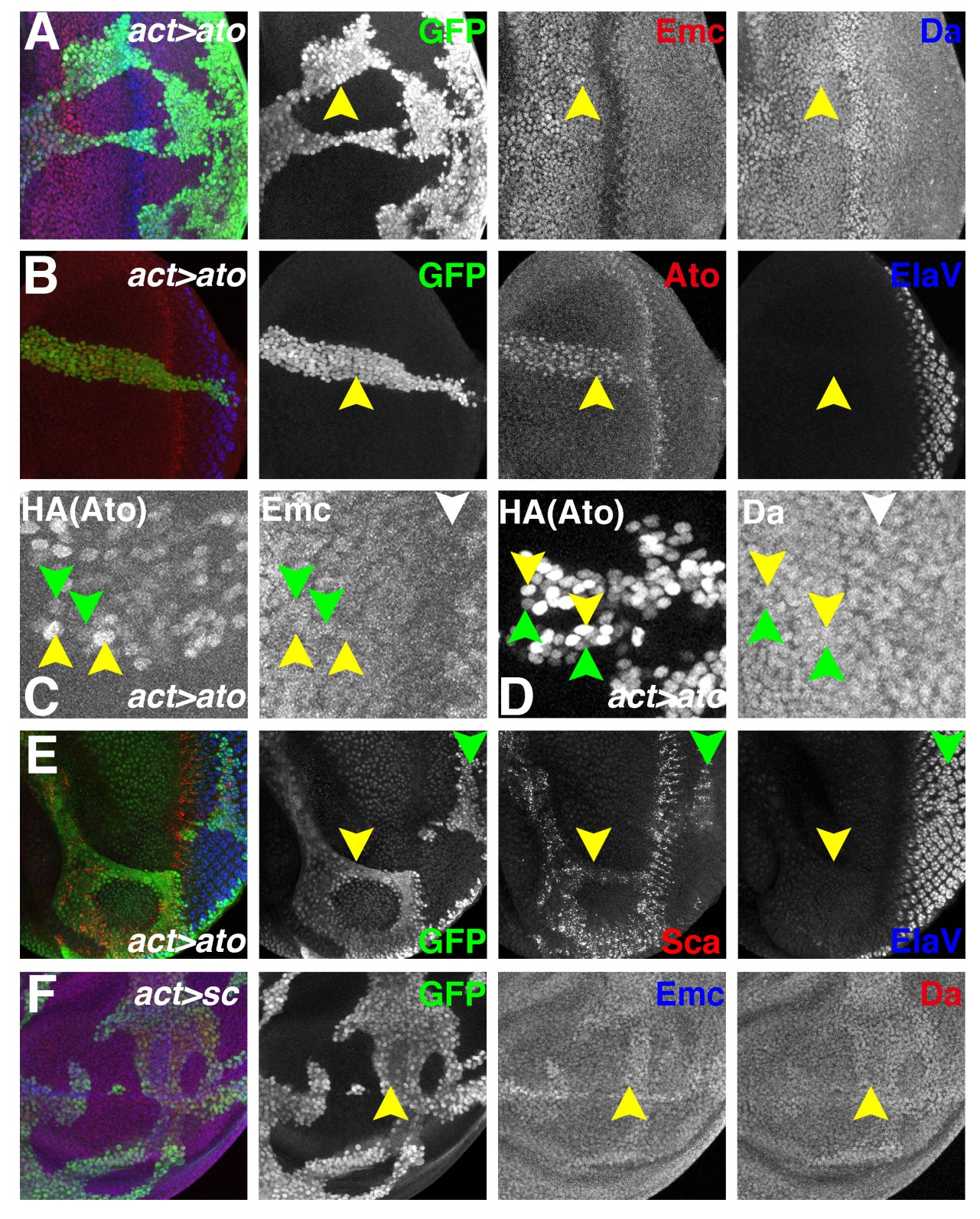

**Figure 7.** Ectopic Ato or Sc does not reduce Emc levels. Flip-on clones expressing *ato* or *sc* using *act-Gal4* and *UAS-ato* or *UAS-sc* lines are marked by GFP (green). (**A**) Ectopic Ato expression from *UAS-ato.ORF-3HA* had little effect on Emc levels (red) but slightly elevated Da (blue). (**B**) Ectopic Ato (red) levels ahead of the furrow were comparable to normal physiological levels in the furrow but failed to induce ectopic neuronal differentiation (Elav: blue). Notably, ectopic Ato levels declined close to the furrow, both anteriorly and posteriorly. (**C**) Cells with higher (yellow arrowheads) and lower (green

*Figure 7 continued on next page*

*Figure 7 continued*

arrowheads) levels of ectopic Ato had similar levels of Emc protein. White arrowhead indicates the morphogenetic furrow. (D) Cells with higher (yellow arrowheads) and lower (green arrowheads) levels of ectopic Ato had similar levels of Da protein. White arrowhead indicates the morphogenetic furrow. (E) Ectopic Ato expression activates its downstream target Sca (red) in the eye disc (arrows) but only affected neuronal differentiation (blue) posterior to the furrow (green arrow). (F) Ectopic Sc expression from *UAS-sc* in wing discs sightly elevated Da (red) in clones and perhaps also Emc (blue) expression. Genotypes: (A–E) *hsFLP; act > CD2>Gal4, UAS-GFP/UAS-ato.ORF-3HA*; (F) *hsFLP; UAS-sc.39/+; act > CD2>Gal4, UAS-GFP/+*.
DOI: https://doi.org/10.7554/eLife.33967.011

The following figure supplements are available for figure 7:

**Figure supplement 1.** Flip-on clones expressing *ato* using *act-Gal4* and various *UAS-ato* lines marked by GFP (green).
DOI: https://doi.org/10.7554/eLife.33967.012

**Figure supplement 2.** Flip-on clones expressing *sc* in eye discs using *act-Gal4* and a *UAS-sc* line are marked by GFP (green).
DOI: https://doi.org/10.7554/eLife.33967.013

(*Figure 7B,E*). In similar experiments, ectopic Sc expression in wing or eye discs only slightly upregulated Da and mildly increased Emc expression (*Figure 7F* and *Figure 7—figure supplement 2*). These results indicated that although proneural genes like Ato and AS-C genes may be required to downregulate Emc, they were not sufficient.

## Uniform emc transcription supports neural patterning

To confirm the primacy of post-translational control of Emc protein, we used the Gal4-UAS system to replace endogenous *emc* expression with ubiquitous transcription under the control of Actin-Gal4 in the background of the embryonic lethal, amorphic genotype $emc^{AP6}/emc^{\Delta1}$. High levels of ubiquitous Emc in the absence of the endogenous locus abolished sensory neurons to various degrees in many tissues (*Figure 8—figure supplement 1A–F*), just like ectopic Emc in the presence of the endogenous locus (*Bhattacharya and Baker, 2011*). At lower temperatures, lower levels of uniform transcription led to different results. Despite uniform transcription, Emc protein patterns resembled wild type (*Figure 8C,F and I*). Emc protein was reduced in the morphogenetic furrow, and higher in regions of the notum primordium, and Da protein levels also resembled the wild type (*Figure 8C,F and I*). One difference, however, was that whereas in wild type higher Emc protein levels were often noticed around the equator near the anterior of the eye disc, a region where *emc* transcription is positively regulated by Notch signaling (*Bhattacharya and Baker, 2009*)(*Figure 1A*), Emc protein levels were uniform here in the flies rescued by uniform *emc* transcription (*Figure 8C*). The rescued *emc* mutants survived to pharate adults, and a small proportion emerged as adults. Both adults and pharate adults exhibited significant rescue of neural patterning. This included almost normal eye development, including the interommatidial bristles (*Figure 8B*), an essentially normal pattern of thoracic macrochaetae, a spaced pattern of some microchaetae (*Figure 8E*), and essentially normal pattern of sensory bristles along the anterior wing margin (*Figure 8H*). Therefore, uniform *emc* transcription was sufficient for most neural patterning, which did not depend critically on patterns of *emc* transcription.

## Discussion

Although proneural gene transcription is highly regulated, uniform proneural transcription still results in a pattern of neurogenesis (*Rodríguez et al., 1990*; *Brand et al., 1993*; *Domínguez and Campuzano, 1993*). A candidate prepattern gene is Emc, a widely-expressed negative regulator of proneural protein function that is down-regulated in proneural regions and neuronal precursor cells (*Cubas and Modolell, 1992*; *Brown et al., 1995*; *Usui et al., 2008*). We, and others, have focused previously on transcriptional regulation of *emc* (*Bhattacharya and Baker, 2009*; *Bhattacharya and Baker, 2011*). Here we report, however, that Emc protein levels were largely determined post-transcriptionally and by its dimerization partners, and also that neuronal patterning was almost normal in the presence of uniform *emc* transcription. Although other proneural prepatterns may exist, Emc is regulated downstream of proneural genes, not upstream. Our findings also suggest that dynamics of HLH protein heterodimer formation and exchange, and ensuing changes in protein stability, may play important roles in neurogenesis and perhaps other processes regulated by bHLH transcription factors (*Figure 9*).

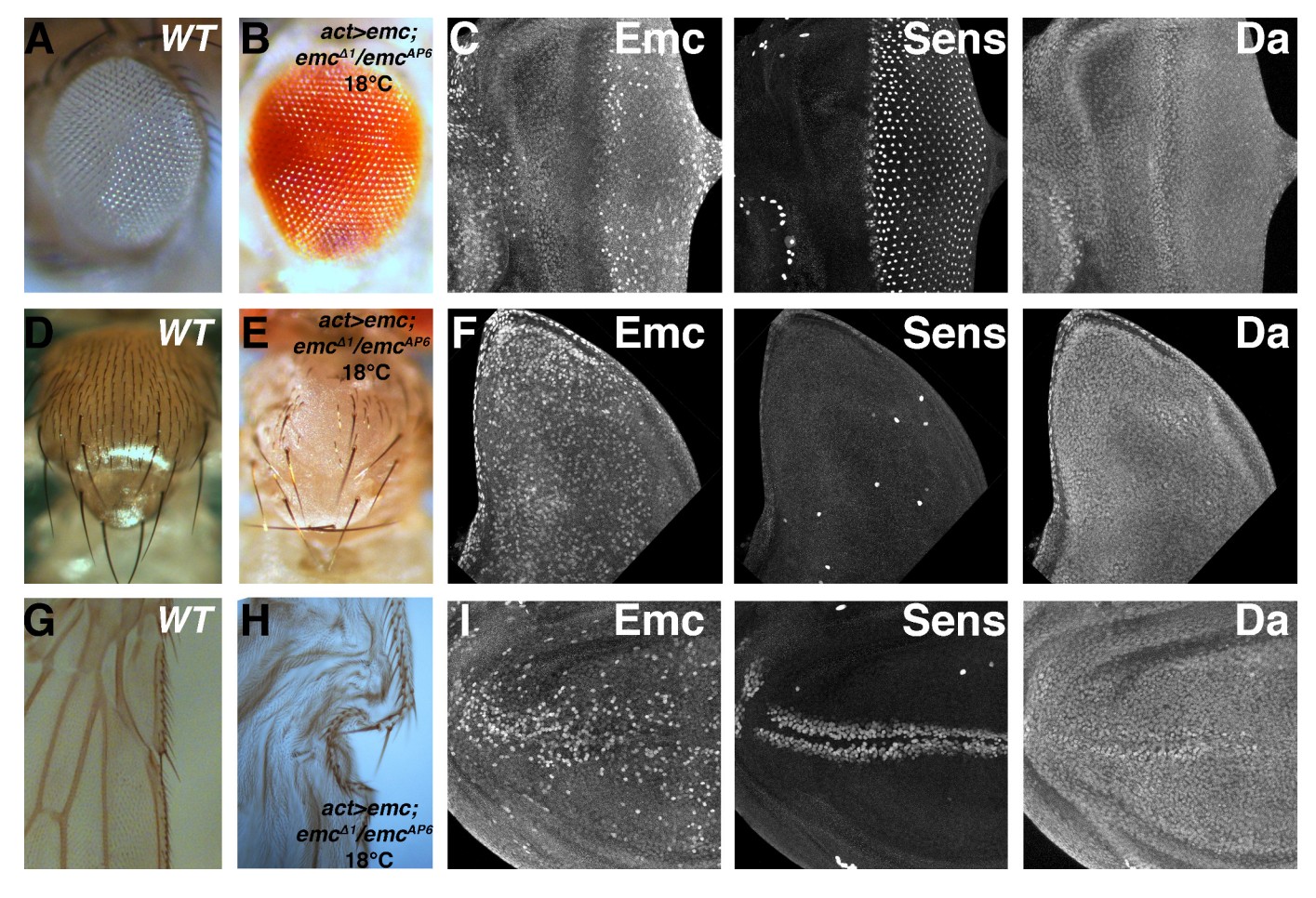

**Figure 8.** Ubiquitous *emc* transcription confers normal expression pattern and neurogenesis. (**A**) Wild type adult eye showing ommatidia and interommatidial bristles. (**B**) Actin-Gal4-mediated ubiquitous transcription of *emc* at 18C in the absence of the endogenous locus gives rise to normal adult eye. (**C**) Eye imaginal discs from the rescued larvae show almost normal protein patterns, including the downregulation of Emc and upregulation of Da in the morphogenetic furrow, and normal neurogenesis shown by Sens staining. Unlike wild type discs, however, Emc protein levels are not discernibly elevated near the equatorial anterior margin (contrast with *Figure 1A*). Scattered cells show higher Emc levels. (**D**) Wild type thorax displays 11 pairs macrochaetae (at least seven pairs are shown here) and evenly-spaced microchaetae. (**E**) Ubiquitous *emc* expression gives rise to nearly all macrochaetae. Spaced microchatae are present over some regions. (**F**) Wing imaginal discs with ubiquitous *emc* expression elevated Emc in many cells. The number and position of Sens positive SOP cells resemble the wild type notum, and they also also showed higher Da expression (compare *Figure 5B*). (**G**) Anterior wing margin from wild type adult flies display mechanosensory and chemosensory bristles. (**H**). Uniform *emc* expression gives rise to normal bristles on the anterior and posterior wing margin (wings from this genotype do not inflate properly). (**I**) Wing imaginal discs from (**H**) show broad Emc expression with higher levels in scattered cells, more frequently in central portions of the wing pouch. Sens and Da shows normal upregulation in the presumptive anterior wing margin. Genotypes: (**A, D, G**) $w^{1118}$; (**B–C, E–F and H–I**) *act > Gal4/UAS-emc 5.3; emc$^{Δ1}$ FRT80/emc$^{AP6}$ FRT80*.

DOI: https://doi.org/10.7554/eLife.33967.014

The following figure supplement is available for figure 8:

**Figure supplement 1.** (A–F) Gal4-driven ubiquitous *emc* transcription at 25C in *emc* mutant.
DOI: https://doi.org/10.7554/eLife.33967.015

Our study first addressed non-proneural regions, where Da levels are uniform and Emc levels are also quite steady. Because *da* is required for Emc expression, while Emc restrains Da expression, their levels might be maintained by homeostatic feedback (*Bhattacharya and Baker, 2011*). The predicted feedback mechanism was not born out by experiment, however. The level of Da expression in fact was not buffered against variation in *da* gene copy number (*Figure 2E,G* and *Figure 2—*

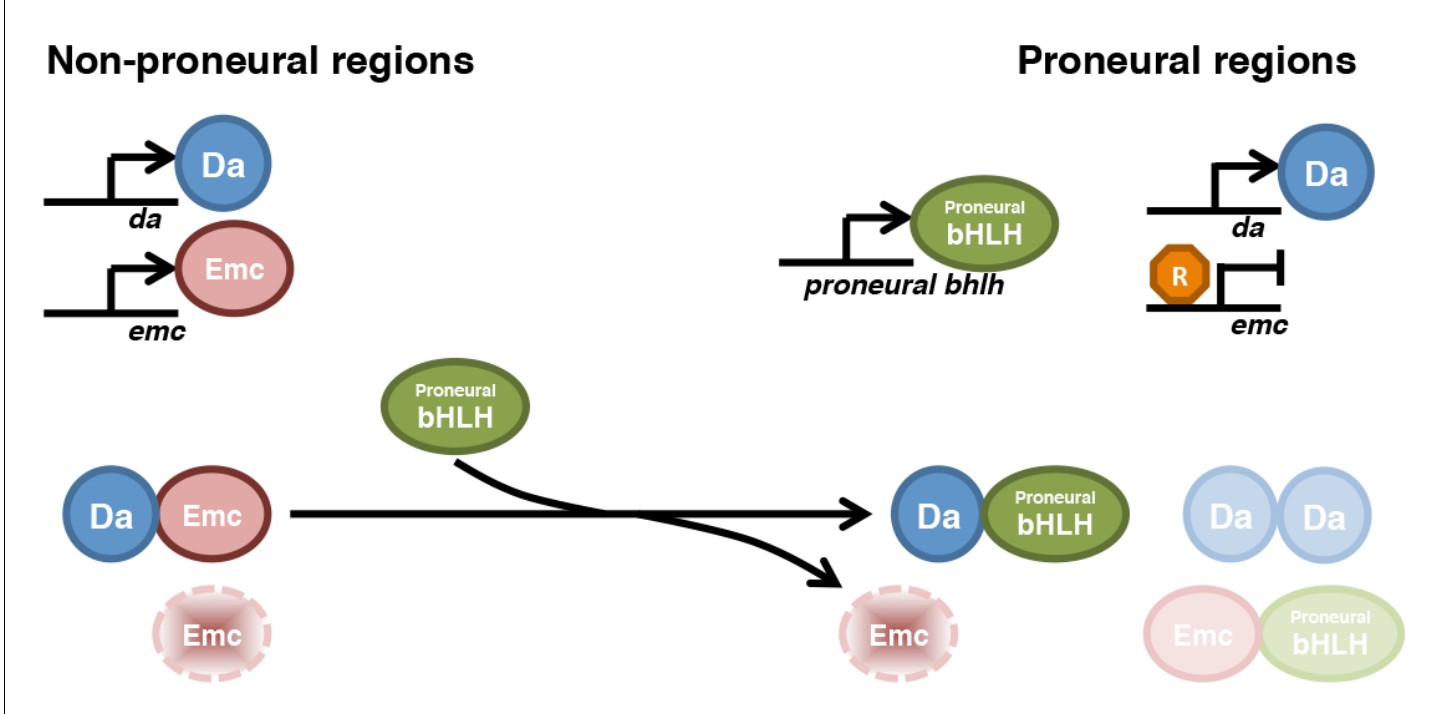

**Figure 9.** Model for HLH protein regulation inside and outside proneural regions. The top part compares gene transcription in most imaginal disc cells (non-proneural) with that in cells in proneural regions. The bottom part compares the protein species active in these cells. In non-proneural regions (left), only *da* and *emc* are transcribed. Emc is short-lived unless dimerized with Da, so that Da levels determine Emc levels. In proneural regions, one or more proneural bHLH genes are activated, and in the morphogenetic furrow *emc* transcription is repressed. Proneural Ato or AS-C proteins bind to Da and are responsible for Emc degradation. Emc may be degraded after displacement from heterodimers. It is possible that Ato (or AS-C) heterodimerizes with Emc, or Da homodimers are also present, and that these species also have distinct stabilities. Our findings show how the changes in Emc and Da levels that are a feature of in all proneural regions depend on post-translational regulation of HLH protein stability.
DOI: https://doi.org/10.7554/eLife.33967.016

*figure supplement 1C*), and not affected by *emc* gene dose or even Emc over-expression (*Figure 2E,H–I* and *Figure 2—figure supplement 1D*). It seems that uniform Da levels outside proneural regions reflect uniform transcription from the simple *da* proximal promoter region, with no contribution from the Da-dependent *da* transcription.

The matching of Emc protein levels with levels of Da led us to discover that Emc protein was short-lived in S2 cells unless Da was co-expressed (*Figure 3E*). We hypothesize that the Emc protein level in vivo is significantly influenced by protein stability and depends on the amount of Da protein available to form heterodimers. Emc and Da proteins may approach 1:1 stoichiometry in imaginal disc cells, with most (if not all) Da and Emc molecules existing as heterodimers. This simple model could explain the lack of evidence for Da-dependent *da* transcription, and the failure of *da* mutations alone to enhance growth, even though *da* can inhibit growth in the absence of *emc* (*Bhattacharya and Baker, 2011*), because Da is heterodimerized with Emc outside of proneural regions (*Figure 9*). Dependence of Emc on Da for stability probably breaks down above a certain threshold, since it is possible to achieve higher Emc levels in over-expression experiments, and for strong Emc over-expression to affect development (*Figure 8—figure supplement 1C–E*) (*Baonza et al., 2000*; *Adam and Montell, 2004*).

The finding that Emc stability depended on Da prompted us to re-evaluate the effects of other potential heterodimer partners. We showed that proneural bHLH proteins were required for reducing Emc levels in the morphogenetic furrow of the eye disc, anterior wing margin, and in SOP primordial of the notum (*Figure 4D,H* and *Figure 5C–D*). Previous studies of the eye had drawn the opposite conclusion, because of studies of an *ato* mutation that encodes a protein expected to lack a functional DNA binding domain but that could contain an intact HLH domain (*Figure 4A*)

(*Bhattacharya and Baker, 2011*). Because changes in Emc and Da levels correlated with expression of an Ato HLH domain, which mediates dimerization, it is likely to be changes in heterodimer partners that altered the levels of Da and Emc proteins in the morphogenetic furrow. We also saw that changes in Emc and Da levels that occur at the anterior wing margin (*Figure 4H–I*) depended on the AS-C, and the same may be true for the SOP cells of the thorax (*Figure 5C*). Another group reported that some aspects of the Emc pattern in the notum primordium were independent of AS-C (*Troost et al., 2015*). Although we confirmed this finding using clones of an AS-C deletion (the $sc^{10.1}$ genotype used before might encode a mutant Sc protein that perhaps could heterodimerize), these patterns reflect mainly regions where Emc is higher than elsewhere. We would describe three levels of Emc in the notum: regions of highest Emc, which do not include proneural regions and where Da expression is not modified; SOP cells, where Emc is reduced and Da elevated; the rest of the notum, where Emc and Da levels are relatively uniform and resemble those of the wing pouch or eye disc. Proneural regions (as defined by Sca-LacZ expression) lie within the latter regions. In our view, the notum region of the wing disc differs from other imaginal discs in that Emc and Da levels are not altered in proneural regions, but only in neural precursor cells themselves, where these changes depend on AS-C. It is worth mentioning that regions of higher Emc are not unique to the notum: the eye disc often has higher Emc along the equatorial region of the anterior eye disc, where it is known that higher *emc* transcription is induced by Notch signaling (*Figure 1A*) (*Bhattacharya and Baker, 2009*).

Despite the importance we demonstrate for proneural proteins in regulating Emc, ectopic Ato or Sc proteins were not sufficient to lower Emc levels prematurely or outside of neurogenic regions (*Figure 7A,F*, *Figure 7—figure supplement 1* and *Figure 7—figure supplement 2*). In proneural regions, proneural proteins may have different properties from the proteins expressed ectopically in other cells. Interestingly, ectopic expression of either Emc or Ato achieved lower protein levels in the vicinity of the morphogenetic furrow (*Figure 2H* and *Figure 7B*).

Our findings challenge the view that Emc levels define a prepattern for neurogenesis sufficient to impose a normal pattern of neurogenesis on a uniform proneural expression pattern, because we found that most variation in Emc levels was downstream of proneural genes. Consistent with this, uniform *emc* transcription was sufficient for most patterns of Emc and Da protein expression and most neural patterning, showing that if Emc contributed to a proneural prepattern, it was not essential for it. This result parallels the earlier discovery that almost normal thoracic neurogenesis can occur in the presence of only uniform AS-C transcription (*Rodríguez et al., 1990*; *Usui et al., 2008*), and suggests that the basis for the neural prepattern could lie elsewhere, for example in post-translational modification of HLH proteins(*Baker and Brown, 2018*). When Emc is expressed ectopically, we do see that Emc protein levels accumulate differently in some locations (*Figure 8*), and this is seen for proneural proteins also, consistent with undescribed factors that determine expression level of these proteins. It can't yet be ruled out, however, that transcriptional regulation of proneural genes and of *emc* each provide redundant patterning information, because we have not investigated whether neural patterning would be normal if *both emc* and proneural genes were transcribed uniformly. In addition, even though the *emc* expression pattern may not be the source of prepattern, Emc may still be a component of the mechanism, since it clearly does suppress neural differentiation at inappropriate locations, which in many cases are locations where ectopic proneural genes seem unable to destabilize Emc.

Like Emc, mammalian ID1, ID2 and ID3 proteins are also short-lived proteins degraded through the ubiquitin proteasome pathway and there is evidence they can be stabilized by heterodimerization with E-proteins (*Deed et al., 1996*; *Bounpheng et al., 1999*; *Lingbeck et al., 2005*). Our studies of the simpler *Drosophila* system indicate that in most cells the E-protein Da is the major single determinant of ID protein stability. It is possible this will be found to be the case in mammalian cells also, which typically express multiple E-proteins and ID-proteins, although it might be necessary to examine cells that lack all mammalian E-proteins. A recent study suggested that Da is made less stable by heterodimerization with Sc, and that Da-Sc heterodimers also affect Enhancer-of-split protein stability and vice versa (*Kiparaki et al., 2015*). Thus, the equilibrium and dynamics of HLH protein dimerization and stability must change when transcription of proneural bHLH genes begins and ends when proneural regions are established and decay, and may also be affected by the Notch signaling that induces expression of bHLH proteins from the E(spl)-C. It would be of interest, in the future, to

investigate the binding properties HLH dimer species and the dynamics of their mixtures more quantitatively than has been done in the past.

# Materials and methods

**Key resources table**

| Reagent type (species) or resource | Designation | Source or reference | Identifiers | Additional information |
|---|---|---|---|---|
| gene (*Drosophila melanogaster*) | *emc* | | FlyBase: FBgn0000575 | |
| gene (*D. melanogaster*) | *da* | | FlyBase: FBgn0267821 | |
| gene (*D. melanogaster*) | ato | | FlyBase: FBgn0010433 | |
| gene (*D. melanogaster*) | sc | | FlyBase: FBgn0004170 | |
| genetic reagent (D. melanogaster) | *da[3]* | PMID: 3802198 | | |
| genetic reagent (D. melanogaster) | *emc[AP6]* | PMID: 7947322 | | |
| genetic reagent (D. melanogaster) | *emc[Δ1]* | this study | | |
| genetic reagent (D. melanogaster) | *act > CD2>Gal4, UAS-GFP* | PMID: 9053304 | | |
| genetic reagent (D. melanogaster) | *ato[1]* | PMID: 8196767 | | |
| genetic reagent (D. melanogaster) | *ato[3]* | PMID: 7635049 | | |
| genetic reagent (D. melanogaster) | *emc-GFP[YB0040]* | PMID: 17179094 | | |
| genetic reagent (D. melanogaster) | *emc-GFP[YB0067]* | PMID: 17179094 | | |
| genetic reagent (D. melanogaster) | *P{PZ}emc[04322]* | PMID: 9529525 | | |
| genetic reagent (D. melanogaster) | *Df(1)sc[B57]* | PMID: 2510998 | | |
| genetic reagent (D. melanogaster) | *Df(1)sc[10-1]* | PMID: 3111716 | | |
| genetic reagent (D. melanogaster) | UAS-HA-da | PMID: 25579975 | | |
| genetic reagent (D. melanogaster) | *UAS-ato.ORF-3HA* | PMID: 23637332 | | |
| genetic reagent (D. melanogaster) | *UAS-sc* | PMID: 8978666 | | |
| genetic reagent (D. melanogaster) | *UAS-ato-4* | PMID: 8324823 | | |
| genetic reagent (D. melanogaster) | *UAS-emc5.3* | PMID: 10804180 | | |
| cell line (D. melanogaster) | S2 | DGRC | Stock Number: 6 | |
| antibody | anti-βGal (mouse) | DSHB | 40-1a | (1:100) |
| antibody | anti-ElaV (rabbit) | DSHB | 7E8A10 | (1:50) |
| antibody | anti-Da (mouse) | PMID: 3802198 | | (1:200) |
| antibody | anti-Emc (rabbit) | Y.N. Jan | | (1:8000) |

*Continued on next page*

*Continued*

| Reagent type (species) or resource | Designation | Source or reference | Identifiers | Additional information |
|---|---|---|---|---|
| antibody | anti-Ato (rabbit) | PMID: 8196767 | | (1: 50000) |
| antibody | anti-Sca (mouse) | PMID: 8622662 | | (1:200) |
| antibody | anti-GFP (rat) | Nacalai Tesque | GF090R | (1:1000) |
| antibody | anti-Sens (guinea pig) | PMID: 10975525 | | (1:50) |
| antibody | anti-V5 (mouse) | Invitrogen | 46–0706 | (1:5000) |
| antibody | anti-Flag (mouse) | Sigma | F3165 | (1:8000) |
| antibody | anti-Tubulin (mouse) | Abcam | ab18251 | (1:5000) |
| antibody | anti-Tubulin (rabbit) | Abcam | ab7291 | (1:5000) |
| antibody | anti-HA (rabbit) | Cell Signaling Tech | C29F4 | (1:1000) |
| antibody | anti-HA (mouse) | Roche | 12CA5 | (1:1000) |
| recombinant DNA reagent | Emc-V5 (plasmid) | this study | | |
| recombinant DNA reagent | Da-Flag (plasmid) | PMID: 25694512 | | |
| recombinant DNA reagent | GFP (plasmid) | PMID: 25694512 | | |

## Mosaic analysis

Mosaic clones were obtained using FLP/FRT mediated mitotic recombination(*Xu and Rubin, 1993*). Larvae were subjected to heat shock for 1 hr at 37°C at 60 ± 12 hr after egg laying, and dissected 72 hr after heat shock. To make 'flip-on' clones, larvae were heat shocked for 30 min instead. All flies were maintained at 25°C unless otherwise stated.

## Drosophila Strains

$w^{1118}$, $da^3$ (*Cronmiller and Cline, 1987*); $emc^{AP6}$(*Ellis, 1994*); $emc^{\Delta1}$(an apparent null allele corresponding to a 1 bp deletion that frameshifts the open reading frame in the 5$^{th}$ codon whose characterization will be described elsewhere); *act > CD2>Gal4, UAS-GFP* (*Pignoni and Zipursky, 1997*), Neufeld, *Neufeld et al., 1998*); *UAS-emc5.3* (*Baonza et al., 2000*); $ato^1$(*Jarman et al., 1994*); $ato^3$(*Jarman et al., 1995*); *UAS-HA-da* (*Wang and Baker, 2015a*); *UAS-ato.ORF-3HA* (*Bischof et al., 2013*); *UAS-sc*(*Parras et al., 1996*); $emc-GFP^{YB0040}$ and $emc-GFP^{YB0067}$ (*Quiñones-Coello et al., 2007*); $P\{PZ\}emc^{04322}$ (*Röttgen et al., 1998*); $Df(1)sc^{B57}$(*González et al., 1989*); *UAS-ato-4* (*Jarman et al., 1993*); $Df(1)sc^{10-1}$ (*Villares and Cabrera, 1987*).

## Immunohistochemistry and image processing

Antibody staining was performed as previously described(*Baker et al., 2014*). The following primary antibodies were used: mouse anti-βGal (1:100, DSHB 40-1a), rabbit anti-βGal, rat anti-ElaV(1:50, DSHB 7E8A10), mouse anti-Da(1:200)(*Cronmiller and Cummings, 1993*), rabbit anti-Emc (1:8000, a gift from Y. N. Jan)(*Brown et al., 1995*), rabbit anti-Ato(1:50000)(*Jarman et al., 1994*), mouse anti-Sca (1:200)(*Lee et al., 1996*), rat anti-GFP(1:1000, Nacalai Tesque GF090R), guinea pig anti-Sens (*Nolo et al., 2000*), mouse anti-HA (1:1000, Roche 12CA5), rabbit anti-HA (1:1000, Cell Signaling Tech C29F4). Seondary antibodies conjugated with Cy2, Cy3 and Cy5 dyes (1:200) were from Jackson ImmunoResearch Laboratories. Multi-labeled samples were sequentially scanned with Leica SP2 or SP5 confocal microscopes. Z-stacks were projected using Max Intensity and processed with ImageJ. Genotypes were identified according to GFP and βGal staining. For quantification of GFP, βGal, Da and Emc levels in mosaic discs, mean fluorescence intensities were measured for all areas of each genotype and averaged for each wing disc. Fluorescence intensities in $gfp^{-/-}$, $lacZ^{-/-}$, $da^{-/-}$ and $emc^{-/-}$ genotypes were measured as an estimate of background to be substracted from anti-GFP, anti- anti-βGal, anti-Da and anti-Emc fluorescence intensities. The wing margin and notum regions were excluded from this analysis of the main wing disc.

## DNA constructs

ORFs of each gene were cloned from cDNA of 0–6 hours $w^{1118}$ embryos to make constructs used in transfection. Emc open reading frame with Kozak sequences were cloned in-frame into pAc5.1/V5-His vector (Invitrogen) to make pAc-Emc-V5 construct. pAc-Da-Flag and pAc-GFP constructs were obtained from was obtained from Dr. Marianthi Kiparaki (*Kiparaki et al., 2015*).

## Cell culture, transient transfection and western blotting

*Drosophila* S2 cells obtained from Drosophila Genomics resource Center were cultured at 25°C in Schneider's Medium supplemented with 10% heat inactivated fetal bovine serum and Penicillin-Streptomycin. Cells were transiently transfected with Effectene Transfection Reagent (Qiagen, Valencia, CA) or TransIT-2020 Transfection Reagent (Mirus, Madison, WI) according to manufacturer's instructions. Cells were treated with 50 μM MG132 or 50 μg/ml cycloheximide where noted (*Kiparaki et al., 2015*). Whole cell lysates were collected 48–72 hr after transfection using RIPA buffer (150 mM sodium chloride, 1.0% NP-40, 0.5% sodium deoxycholate, 0.1% SDS and 50 mM Tris, pH 8.0) with addition of protease inhibitors cocktail (Roche) and phosphatase inhibitors cocktails (Sigma). Total protein concentration was determined using Pierce BCA Protein Assay Kit (ThermoFisher Scientific). Protein lysates were separated on 10–12% homemade SDS–polyacrylamide gels and electrotransferred onto PVDF membranes (Bio-Rad) for following detection by western blotting. The following primary antibodies were used for western blotting: mouse anti-V5 (1:5000, Invitrogen 46–0706), mouse anti-Flag (1:8000, Sigma F3165), mouse anti-Tubulin (1:5000, Abcam ab18251), rabbit anti-Tubulin (1:5000, Abcam ab7291). Secondary antibodies conjugated with IRDye 680RD and IRDye 800CW were used (LI-COR, Lincoln, NE). Membranes were imaged on LI-COR Odyssey scanner and images were quantified in ImageJ.

## Sequencing of ato mutant alleles

Both $ato^1$ and $ato^3$ flies were outcrossed to $w^{1118}$ flies to obtain $ato^1/+$ and $ato^3/+$ flies. Genomic DNA was isolated from $w^{1118}$, $ato^1/+$ and $ato^3/+$ flies and PCR products were obtained using primers flanking the endogenous *ato* locus. Amplified products were gel purified and subjected to Sanger sequencing. Re-sequencing of $ato^1$ confirmed three point mutations (8278198G > A, 8278884G > T and 8278907A > T, numbers represented genomic coordinates on chromosome 3L).

## Acknowledgements

We thank Drs. Abhishek Bhattacharya, Jorge Blanco, Jean Hebert, Andreas Jenny, Marianthi Kiparaki, Ertugrul Ozbudak, Francesca Pignoni and Lan-Hsin Wang for comments on the manuscript, Dr. Abhishek Bhattacharya for initial contributions to the project, and Dr. Marianthi Kiparaki for DNA constructs. *Drosophila* stocks were obtained from the Flytrap Project, the Zurich ORFeome Project (FlyORF) and the Bloomington *Drosophila* Stock Center (supported by NIH P40OD018537). S2 cells were obtained from the *Drosophila* Genomics Resource Center (supported by NIH 2P40OD010949-10A1). Confocal microscopy was performed in the Analytical Imaging Facility of the Albert Einstein College of Medicine (supported by the NCI P30CA013330). DNA sequencing was performed by the Genomics Core of Albert Einstein College of Medicine. This work was supported by the NIH grant GM047892. Data in this paper are from a thesis submitted in partial fulfillment of the requirement for the degree of Doctor of Philosophy in the Graduate Division of Biomedical Sciences, Albert Einstein College of Medicine, Yeshiva University, USA.

## Additional information

### Funding

| Funder | Grant reference number | Author |
| --- | --- | --- |
| National Institute of General Medical Sciences | GM047892 | Nicholas E Baker |

The funders had no role in study design, data collection and interpretation, or the decision to submit the work for publication.

## Author contributions

Ke Li, Conceptualization, Data curation, Formal analysis, Validation, Investigation, Visualization, Methodology, Writing—original draft, Writing—review and editing; Nicholas E Baker, Conceptualization, Supervision, Funding acquisition, Writing—review and editing

## Author ORCIDs

Ke Li ⓘ https://orcid.org/0000-0002-0737-1045
Nicholas E Baker ⓘ http://orcid.org/0000-0002-4250-3488

## Decision letter and Author response

Decision letter https://doi.org/10.7554/eLife.33967.019
Author response https://doi.org/10.7554/eLife.33967.020

# Additional files

## Supplementary files

• Transparent reporting form
DOI: https://doi.org/10.7554/eLife.33967.017

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
