## [Decision Letter]

[Editors’ note: a previous version of this study was rejected after peer review, but the authors submitted for reconsideration. The first decision letter after peer review is shown below.]

Thank you for submitting your work entitled "Regulation of the *Drosophila* ID protein Extra macrochaetae by proneural dimerization partners" for consideration by *eLife*. Your article has been reviewed by three peer reviewers, and the evaluation has been overseen by a Reviewing Editor and a Senior Editor.

Our decision has been reached after consultation between the reviewers. Based on these discussions and the individual reviews below, we regret to inform you that your work will not be considered further for publication in *eLife* as the findings at this stage.

Specifically, two of the three reviewers, who are all experts in your field, request a significant number of additional experiments to bolster your conclusions. The data and hypothesis have potential, but it is most likely that it will take more than two months to address the concerns and we can therefore not accept the manuscript. Furthermore, from some of the current inconsistencies between data and text pointed out in all three reviews, the consultations suggest that the suggested work may well show far more complexity to the regulatory network that will require a significantly more fine-tuned model. We suggest that you carefully read the reviews and determine if you are willing and able to address these concerns. If you can detail that this is indeed feasible and decide to submit a manuscript to *eLife* rather than elsewhere, we will be pleased to afresh consider an extensively edited and revised manuscript in which you address all key concerns and decide if it should be reviewed.

Reviewer #1:

Li and Baker report a very interesting set of observations suggesting complex post-transcriptional, and very likely post-translational cross-regulation between Proneural Proteins (Ato, Ac-Sc), E-Proteins (Da) and ID proteins (Emc) in *Drosophila*. Building upon previous work by the same group, the authors uncover evidence consistent with the idea that heterodimer formation between Emc and Da may compete with heterodimer formation between Da and Ato/Ac-Sc and that this competition to a significant degree determines Emc levels and thus potentially neurogenic regions in the fly PNS.

While the idea of a critical role for post-translational regulation of key proteins in neurogenesis, in this case that of ID proteins, is very interesting and consistent with a growing body of evidence supporting this notion, this study does not provide sufficient evidence for the mechanisms of such regulation and its impact on neurogenesis. Much more work is needed to consolidate the findings, extend them to a level that allows the extraction of a model and understand their consequences for cell fate transitions in the nervous system.

1) The genetic evidence presented is largely consistent with the model, but not entirely. In subsection “Da and Emc protein levels are proportional to da gene dose” it is stated that the evidence in Figure 2 argues against homeostatic feedback regulation because "In the background wild type for emc (i.e. emc+/+), cells with two copies of the wild type da gene had almost twice as much Da protein as cells with only one copy (Figure 2M)". Looking at Figure 2M, this is clearly not the case. The levels of Da protein are far less than double. In contrast in emc-/- cells the levels of Da are more than double the gene dosage. This, in fact, is precisely consistent with homeostatic feedback. On the same page, it is stated, correctly, that Da levels are indistinguishable between emc-/+ and emc+/+. Again, this seems precisely what one would predict from homeostasis: that it buffers against minor changes in the dosage of the regulator. Thus, the very notion the study start with discounting seems to be supported by, rather than contradicted by, this particular set of data.

2) Aside from measuring steady state protein levels in S2 cells, the paper lacks biochemical evidence for the model, particularly for competition between Emc and Proneural Proteins for forming heterodimers with Da. More biochemical and and genetic evidence is needed to demonstrate this model. For example, there are no in vitro competition experiments performed. Another example: the difference in the effect of the two Ato alleles (Ato1 vs Ato3) is intriguing and an important piece of data. It is consistent with the idea that the bHLH domain in needed as opposed to the transcriptional activity, but Ato3 lacks much more than the bHLH domain. A more careful dissection is needed.

3) There is clear evidence in the paper for a complex set of regulatory interactions, because over expression of Ato has no effect on Emc levels and because there is clearly some level of transcriptional control. The paper however provides no data to explain these effects, and how both transcriptional and post-transcriptional mechanisms interact to ensure the precision of the Emc pattern.

4) What are the implications for neurogenesis for interfering with the post-translational regulation of Emc?

Reviewer #2:

This paper addresses the role of Emc (Id in mammals) in patterning proneural domains in *Drosophila*. Earlier studies have indicated that Emc provides a pre-pattern that is decoded by Da to direct the pattern of proneural gene expression. The proposed decoding mechanism involved positive regulation of proneural factors by Da, titration (inhibition) of Da by Emc and Da auto-regulation. The data presented here challenges this view.

First, the authors nicely show that the physiological levels of Emc proteins are set by the level of Da proteins (Figure 2; this effect does not seem to be mediated by a transcriptional regulation of emc by Da, Figure 7) and that Emc is stabilized by Da in S2 cells (Figure 3). They further show that the presence of a mutant/inactive Ato protein is sufficient to up-regulate Da, possibly via a stabilization mechanism (not further investigated here), and to down-regulate Emc (Figure 4). How Ato down-regulates Emc remains, however, unclear. Based on the findings reported in Figure 2 and Figure 3, a possible mechanism would be a competition for the binding of Da, resulting in the destabilization of free Emc. However, since ectopic Ato (or Sc) is not sufficient to down-regulate Emc (Figure 5), this mechanism is not favored. Another possibility would be that Ato (or Sc) represses the transcription of the emc gene. However, Ato is not required to repress emc transcription in the morphogenetic furrow. Thus, how Emc is down-regulated in proneural domains at the mRNA and protein levels remain unclear. Also, the functional significance of this down-regulation for proneural patterning is not clear (more generally, whether the patterned expression of emc is relevant for its function remains to be addressed; see below). Finally, the authors suggest that Da is required to activate (or at least maintain) emc gene expression, but only in a region close to/ahead of the furrow in the eye disc (Figure 7; emc is then repressed in the furrow; and Da does not seem to be required in wings).

In summary, this manuscript reports one interesting finding i.e. Da stabilizes Emc and sets Emc protein levels. However, the extent to which this mechanism contributes to the endogenous Emc protein pattern is not entirely clear to me. This manuscript also reports a new model (Figure 8) suggesting that the pattern of Ato/Sc is established in response to positional information independently of the Emc pattern and that Emc protein levels are merely a read-out of proneural activity. In this model, Emc appears to merely keep Da inactive in the absence of proneural factors. This model is supported by the data but remains largely untested.

Essential revisions:

1) To further test the relevance (or lack thereof) of the expression pattern of the emc gene, the authors should test whether ubiquitous expression of emc (at the gene level; i.e. ubi-emc in emc mutants) results in a normal Emc protein pattern and provides proper emc activity. Also, the model predicts that changes in the level of expression of the emc gene should have no significant effect on proneural patterning. This should be tested.

2) A potential limitation of this study is that it challenges a view proposed by several groups (Modolell, Simpson, Klein) who worked on a slightly different context (proneural clusters of the dorsal thorax). It could be appropriate to revisit this view also in this context (or at least justify why conclusions obtained in the eye necessarily applies to another developmental context).

Reviewer #3:

Li and Baker present a thorough analysis of the interdependence of the expression levels of two interacting HLH proteins, Da (a DNA-binding activator) and Emc (a non-DNA-binding inhibitor of Da). Earlier work from the same group had shown that Da activates emc expression, which then feeds back to repress da transcription by preventing Da from activating a da autoregulatory enhancer (Bhattacharya et al., 2011). The complex Emc expression pattern was thought to provide a prepattern for neural differentiation, in the sense that Da/proneural activators would be more active in regions of lowest emc expression. The authors claim that emc does not set the prepattern for neurogenesis, but instead its complex expression pattern depends to a great extent on the activity of Da and its proneural patterns.

The crucial findings of Li and Baker are:

1) Da protein levels are proportional to the da gene dosage in the range of 0-2. However, Emc protein levels do not similarly reflect the emc gene dosage (range of 0-2), instead they also reflect the da gene dosage! Da levels are not affected by emc gene dosage in the range of 1-2, but are greatly upregulated in emc null clones, due to the above-mentioned release of inhibition.

2) The unexpected dependence of Emc protein levels on Da protein levels probably comes from the dramatic stabilization of Emc by Da, as determined by half-life measurements.

3) The upregulation of Da and concomitant downregulation of Emc seen in proneural regions of the eye and wing margin need the expression of proneural bHLH proteins with intact HLH domains, but not necessarily DNA binding activity. This suggests that a proneural partner displaces Emc from Da, which leads to degradation of Emc and auto-activation of Da. Some unknown factor must also contribute to this process, since ectopic expression of proneural genes in non-proneural regions cannot recapitulate this effect on Emc and Da levels.

4) emc transcription is not affected by ato or da mutations (exception: there is a moderate effect of da mutations only in the posterior part of the eye disk).

This three-way post-transcriptional interplay among Da, Emc and proneurals represents an important in vivo demonstration of properties of HLH proteins that had only been shown in vitro. It also paves the way for deciphering the biological consequences of these important HLH heterodimerization properties, adding the interesting twist of stability regulation. That said, I have a few concerns that I will list below:

a) The authors focus their "narrative" on how Emc does not represent a proneural prepattern, but rather reflects a proneural "response". I would say that it is both. They themselves show that the complex RNA pattern of emc does not depend (much) on Da or proneural factors, so this is patterned by something else. Granted, this complex pattern is greatly "evened out" at the protein level, thanks to the mutual interactions described by the authors. Still some proneural cluster "lows" in Emc protein (e.g. the morphogenetic furrow) have a clear proneural-independent transcriptional component, so they are expected to predispose these areas to become neurogenic, namely act as a prepattern. I would change the narrative in such a way to clarify that emc contributes to the prepattern (it cannot be the entire prepattern, since there is still some patterning of bristles in the absence of emc), but at the same time it is itself modulated by proneural activity (at the protein level). After all, whether Emc is or is not a prepatterning factor seems of lesser importance than the dissection of its molecular mode of action.

b) They should point out that the previously reported activation of emc transcription by da was a misinterpretation of the destabilization of Emc in da null clones and that instead Da only affects transcription of emc behind the morphogenetic furrow (and that only modestly, as shown in Figure 7—figure supplement 1).

c) They should point out that the dependence of Emc on Da levels does break down after a certain threshold. Overexpression of emc in Figure 2O-R causes an enormous increase of Emc protein without concomitantly increasing Da. Probably the degradation machinery of Emc is saturated. Intriguingly Da may also play some role even at these unnaturally high Emc levels, since in the furrow, where Da is engaged by Ato, this excess Emc seems to accumulate less. Therefore, its high accumulation in non-proneural regions is not solely because of saturation of the degradation machinery, since the latter seems to work better in the furrow.

d) The language needs to be improved. There are many places where the text is hard to follow and the meaning difficult to extract. Probably, with a thorough rewriting, the scientific points (a-c) above will also become clearer.

[Editors’ note: what now follows is the decision letter after the authors submitted for further consideration.]

Thank you for submitting your article "Regulation of the *Drosophila* ID protein Extra macrochaetae by proneural dimerization partners" for consideration by *eLife*. Your article has been reviewed by two peer reviewers, and the evaluation has been overseen by a Reviewing Editor, Hugo Bellen, and K VijayRaghavan as the Senior Editor. The reviewers have opted to remain anonymous.

The reviewers have discussed the reviews with one another and the Reviewing Editor has drafted this decision to help you prepare a revised submission..

Reviewer #1:

The revised manuscript is much improved, especially as it is more focused on the main message. The new experiment showing that uniform moderate levels of EMC driven by an exogenous promoter rescue neurogenesis certainly support the conclusion that post-transcriptional modulation of EMC is key to patterning neurogenesis. The new data showing that Da protein levels increase linearly with gene dosage are more convincing. Finally, the in vivo data showing that availability of Da regulates EMC protein levels are very nice.

What remains rather circumstantially substantiated in my view is their mechanistic model of the regulation of EMC protein levels by Proneural proteins by competition for Da. I agree with the authors that the few data they have in this regard do not contradict this notion, and I in fact like the idea mostly because it fits the strong response of EMC to Da levels within the proneural domain under wild type conditions. I am surprised that they were unable to express Atonal in S2 cells to test their model more directly because expression of Ato in vitro has been reported previously.

The idea that the proneural pre-pattern depends on Da and Proneural proteins, not ID proteins is novel and interesting. Furthermore, it fits nicely with the emerging paradigm that post-transcriptional/translational modification of bHLH proteins is a key determinant of neurogenesis. The authors could do a better job referencing that literature in their Discussion section.

In summary, the authors' model is the simplest and most parsimonious explanation of their data, but their advance of a specific mechanism for their observations remains relatively weak.

Reviewer #2:

This paper challenges the commonly held view that the ID factor Emc acts as a prepattern regulator of proneural activity during neurogenesis in *Drosophila*. Emc is known to heterodimerize with the E protein Da, thereby blocking its DNA-binding activity, hence its ability to regulate proneural gene expression. This paper reports on an alternative model whereby Da would stabilize Emc in the absence of proneural factor and proneural factors, patterned independently of Emc activity, would negatively regulate this stabilizing activity of Da towards Emc to generate the observed pattern of Emc protein accumulation.

Several interesting observations support this model. First, in contexts of low proneural activity (S2 cells, wing pouch), the accumulation levels of Emc proteins appear to depend on Da levels. The analysis of Emc and Da levels in da, emc clones is really nice (Figure 2) and the stabilization of Emc by Da in S2 cells is clear (Figure 3).

Second, in the proneural region of the eye, Ato is required for the accumulation of Da and the down-regulation of Emc in neural cells (Figure 4). The analysis of the ato1 and ato3 mutations is convincing. The observation that Emc is stabilized in the absence of Ato (Figure 4D and Figure 6) is interesting. However, how Ato destabilizes Emc is not understood: Ato does not appear to be sufficient to destabilize Emc (Figure 7). It is also not clear whether Ato acts indirectly via Da.

Third, non-patterned expression of the emc gene largely rescues the emc mutant phenotype (Figure 8), indicating that Emc transcription does not provide key patterning cues.

In summary, the strength of this paper is that it convincingly shows that the current model needs to be revised. Further experiments are, however, needed to strengthen the alternative model proposed here.

Essential revisions:

1) It would be nice to confirm the changes seen in Da levels by IF using Western blots, for instance by comparing Da levels in emc+/+ vs emc+/- discs (and Emc in da+/+ vs da+/- tissues). Please test.

2) Figure 2E: it is not clear whether the increased Da levels seen in emc mutant cells result from protein stabilization (indicative of Emc targeting Da for degradation, as suggested by Figure 3F) or from increased transcription (due for instance to ectopic/premature expression of proneural factors). Please clarify.

3) Figure 3G: please control that nub>da does not affect emc gene expression (in situ, enhancer-trap…)

4) Subsection “Ato is required for altering Da and Emc levels in the morphogenetic furrow”. The conclusion that Da up-regulation is transcriptional and bHLH-mediated needs to be supported by experimental evidence. Could the authors exclude the possibility that Emc also targets Da for degradation (see Figure 3F)?

5) Subsection “AS-C is required for altering Da and Emc levels in wing disc proneural regions”: This conclusion is not convincing. I find it hard to compare Da levels based on different IF experiments. Clonal analysis is required.

6) Subsection “AS-C is required for altering Da and Emc levels in wing disc proneural regions”: Whether AS-C regulates Emc/Da levels in sensory cells is difficult to study in the absence of these cells. The data shown in Figure 5B-D are not convincing.

7) Figure 7B: heterogenous levels of Ato appear to be detected in the non-proneural region (Figure 7B). Therefore, a per-cell quantification of the relative Emc/Da levels is needed (Figure 7A). It would be nice to test whether Ato levels correlate with Emc/Da levels. Also, what is causing the observed heterogeneity in Ato levels?

8) Figure 7—figure supplement 1A: the clone of act>ato cells (right) exhibits high Emc and high Da levels. This seems to contradict the model whereby Ato competes Emc for binding Da. Please comment.

9) Figure 8: what is the phenotype of emcD1/emcAP6 flies? are these mRNA null? what is the emc mRNA pattern in the mutant and rescued eye discs? Is the latter similar to the protein pattern shown in 8C? does the UAS-ems construct contain the 3'UTR of the emc gene? Does it matter? Also, is the emc mutant combination used in Figure 8 protein null?

---

## [Author Response]

[Editors’ note: the author responses to the first round of peer review follow.]

Specifically, two of the three reviewers, who are all experts in your field, request a significant number of additional experiments to bolster your conclusions. The data and hypothesis have potential, but it is most likely that it will take more than two months to address the concerns and we can therefore not accept the manuscript. Furthermore, from some of the current inconsistencies between data and text pointed out in all three reviews, the consultations suggest that the suggested work may well show far more complexity to the regulatory network that will require a significantly more fine-tuned model. We suggest that you carefully read the reviews and determine if you are willing and able to address these concerns. If you can detail that this is indeed feasible and decide to submit a manuscript to eLife rather than elsewhere, we will be pleased to afresh consider an extensively edited and revised manuscript in which you address all key concerns and decide if it should be reviewed.In addition to a complete rewrite of the manuscript, we have added three additional experiments requested by reviewers and which strengthen the paper: (1) We extended the analysis of Da protein levels in *da* gene doses from 0-2 to 0-4; (2) We showed that uniform transcription of *emc* is sufficient to rescue nearly all gene expression patterns and most of the *emc* mutant phenotype; (3) We explored the regulation of Emc and Da in the notum region of the thorax and explained how our findings are related to previously-published models.Reviewer #1:Li and Baker report a very interesting set of observations suggesting complex post-transcriptional, and very likely post-translational cross-regulation between Proneural Proteins (Ato, Ac-Sc), E-Proteins (Da) and ID proteins (Emc) in Drosophila. Building upon previous work by the same group, the authors uncover evidence consistent with the idea that heterodimer formation between Emc and Da may compete with heterodimer formation between Da and Ato/Ac-Sc and that this competition to a significant degree determines Emc levels and thus potentially neurogenic regions in the fly PNS.While the idea of a critical role for post-translational regulation of key proteins in neurogenesis, in this case that of ID proteins, is very interesting and consistent with a growing body of evidence supporting this notion, this study does not provide sufficient evidence for the mechanisms of such regulation and its impact on neurogenesis. Much more work is needed to consolidate the findings, extend them to a level that allows the extraction of a model and understand their consequences for cell fate transitions in the nervous system.1) The genetic evidence presented is largely consistent with the model, but not entirely. On page 8 it is stated that the evidence in Figure 2 argues against homeostatic feedback regulation because "In the background wild type for emc (i.e. emc+/+), cells with two copies of the wild type da gene had almost twice as much Da protein as cells with only one copy (Figure 2M)". Looking at Figure 2M, this is clearly not the case. The levels of Da protein are far less than double. In contrast in emc-/- cells the levels of Da are more than double the gene dosage. This, in fact, is precisely consistent with homeostatic feedback. On the same page, it is stated, correctly, that Da levels are indistinguishable between emc-/+ and emc+/+. Again, this seems precisely what one would predict from homeostasis: that it buffers against minor changes in the dosage of the regulator.Thus, the very notion the study start with discounting seems to be supported by, rather than contradicted by, this particular set of data.

Reviewer 1 complained that Figure 2M (now Figure 2E in the revised submission) shows that the da+/+ genotype encodes less than twice the amount of Da protein as the da(+/-) genotype, supporting the idea of feedback regulation of Da levels rather than discrediting it.

There are multiple responses to this criticism.

The First is that although the reviewer is correct about the relative values of these particular two datapoints, their conclusion is not supported by the rest of the quantification in this and related Figures. We have now measured the relative levels of Da protein in da[+/+] versus da[+/-] cells in 4 independent experiments in the emc[+/+] genotype (Figure 2E, Figure 2—figure supplement 1C, and twice in Figure 2G) as well as once in the emc[+/-] genotype (in Figure 2E). In all the other cases the Da protein level was clearly doubled by doubled Da gene dose. In addition, 3x and 4x Da gene dose has now been added (Figure 2G) and also look linear. In addition, it is not the level for da[+/+] in Figure 2E that was unexpectedly low, it is that the level for da[+/-] was anomalously high. Although the da[+/-] emc[+/+] datapoint in Figure 2E is a little higher than in the other da[+/-] examples, it is not statistically different from them. The same graph shows that the Da level in da[+/ +] is identical to the Da level in da[+/+] emc[+/-]. In fact, we suspect the reviewer might have missed the latter datapoint because it is partially covered by the former and have changed the format of the graphs to help make this clearer. These da[+/+] genotypes also have indistinguishable Emc protein levels (Figure 1F) so how can the identical Da levels be evidence for feedback? It is unfortunate that the First graph of Da protein levels encountered is this one that contains an outlier, and we experimented with different orders of presentation, or moving Supplemental figures into the main Figure, but it is just more logical for the argument to present Figure 2E First, so we have kept the order of presentation and added a comment to the Figure legend.

The second response is to emphasize that while these gene dose experiments certainly generated our hypothesis, this was then tested in a very stringent assay when we over-expressed Emc without seeing any diminution in Da levels in nonproneural cells (Figures 2H,I). This lack of effect of even very high Emc levels is inconsistent with the notion of continuous transcriptional feedback that maintains constant Da levels (please see subsection “Da and Emc protein levels are proportional to da gene dose”).

*2) Aside from measuring steady state protein levels in S2 cells, the paper lacks biochemical evidence for the model, particularly for competition between Emc and Proneural Proteins for forming heterodimers with Da. More biochemical and and genetic evidence is needed to demonstrate this model. For example, there are no* in vitro *competition experiments performed. Another example: the difference in the effect of the two Ato alleles (Ato1 vs Ato3) is intriguing and an important piece of data. It is consistent with the idea that the bHLH domain in needed as opposed to the transcriptional activity, but Ato3 lacks much more than the bHLH domain. A more careful dissection is needed.*

We acknowledge that we have not directly shown that Emc and proneural proteins compete for Da in vitro. Such an experiment is not trivial to do with EMSA assays. Because neither Emc/Da nor Emc/Ato heterodimers would bind DNA, only Ato/Da heterodimers could be quantified directly. Because Ato does not bind DNA except as a Ato/Da heterodimer, whether Emc disrupts DNA binding through interaction with Ato or with Da would have to be inferred indirectly. Our lab is considering measuring the interactions directly by biophysical means e.g. analytical ultracentrifugation, but these experiments will be more time-consuming. We also have to take account of what projects can gain funding. Because the significance of directly measuring the affinities of various HLH proteins is not yet generally apparent (perhaps because our work is still unpublished), at present it is a common opinion among grant reviewers that such experiments represent an incremental increase in knowledge of little practical significance. In this regard we state that in vitro data like that suggested by the reviewer would provide further support to our model (Discussion section). We suggest that our paper should stimulate more interest in these questions.

3) There is clear evidence in the paper for a complex set of regulatory interactions, because over expression of Ato has no effect on Emc levels and because there is clearly some level of transcriptional control. The paper however provides no data to explain these effects, and how both transcriptional and post-transcriptional mechanisms interact to ensure the precision of the Emc pattern.

It is clear from several reviewers’ comments that data on the transcriptional regulation of Emc was complicating while adding little to the manuscript and we have removed most of this data. We think the revised manuscript is more focused as a result.

4) What are the implications for neurogenesis for interfering with the post-translational regulation of Emc?

The consequences of preventing Emc degradation for both Da expression and for neural patterning were described in detail previously (Bhattacharya and Baker, 2011).

The best experiment was one in which enGal4 at 23°C drove a level of Emc expression that was not detectable outside of proneural regions but allowed Emc protein to persist in neural cells. This led to loss of ~60% of macrochaetae (Figures 6E-J and S5C-H of Bhattacharya and Baker, 2011.

Reviewer #2:This paper addresses the role of Emc (Id in mammals) in patterning proneural domains in Drosophila. Earlier studies have indicated that Emc provides a pre-pattern that is decoded by Da to direct the pattern of proneural gene expression. The proposed decoding mechanism involved positive regulation of proneural factors by Da, titration (inhibition) of Da by Emc and Da auto-regulation. The data presented here challenges this view.First, the authors nicely show that the physiological levels of Emc proteins are set by the level of Da proteins (Figure 2; this effect does not seem to be mediated by a transcriptional regulation of emc by Da, Figure 7) and that Emc is stabilized by Da in S2 cells (Figure 3). They further show that the presence of a mutant/inactive Ato protein is sufficient to up-regulate Da, possibly via a stabilization mechanism (not further investigated here), and to down-regulate Emc (Figure 4). How Ato down-regulates Emc remains, however, unclear. Based on the findings reported in Figure 2 and Figure 3, a possible mechanism would be a competition for the binding of Da, resulting in the destabilization of free Emc. However, since ectopic Ato (or Sc) is not sufficient to down-regulate Emc (Figure 5), this mechanism is not favored. Another possibility would be that Ato (or Sc) represses the transcription of the emc gene. However, Ato is not required to repress emc transcription in the morphogenetic furrow. Thus, how Emc is down-regulated in proneural domains at the mRNA and protein levels remain unclear. Also, the functional significance of this down-regulation for proneural patterning is not clear (more generally, whether the patterned expression of emc is relevant for its function remains to be addressed; see below). Finally, the authors suggest that Da is required to activate (or at least maintain) emc gene expression, but only in a region close to/ahead of the furrow in the eye disc (Figure 7; emc is then repressed in the furrow; and Da does not seem to be required in wings).In summary, this manuscript reports one interesting finding, i.e. Da stabilizes Emc and sets Emc protein levels. However, the extent to which this mechanism contributes to the endogenous Emc protein pattern is not entirely clear to me. This manuscript also reports a new model (Figure 8) suggesting that the pattern of Ato/Sc is established in response to positional information independently of the Emc pattern and that Emc protein levels are merely a read-out of proneural activity. In this model, Emc appears to merely keep Da inactive in the absence of proneural factors. This model is supported by the data but remains largely untested.

Reviewer 2 states the manuscript contains one interesting finding, that Da stabilizes Emc levels. The revised manuscript also shows that; (a) proneural genes are required to destabilized Emc; (b) uniform Emc transcription is sufFicient for largely normal patterning. Therefore, we hope the reviewer finds the revised manuscript more substantial.

Essential revisions:1) To further test the relevance (or lack thereof) of the expression pattern of the emc gene, the authors should test whether ubiquitous expression of emc (at the gene level; i.e. ubi-emc in emc mutants) results in a normal Emc protein pattern and provides proper emc activity. Also, the model predicts that changes in the level of expression of the emc gene should have no significant effect on proneural patterning. This should be tested.

As suggested, we have now tested whether ubiquitous Emc transcription can rescue the patterning of Da and Emc proteins and the pattern of neurogenesis and confirmed that it does. This contributes significantly to the significance of the study by showing that the Emc transcription pattern is not essential for the neural prepattern as previously assumed (subsection “Proneural genes are not sufficient to regulate Da or Emc protein levels.”; subsection “Uniform Emc transcription supports neural patterning”; Figure 8).

2) A potential limitation of this study is that it challenges a view proposed by several groups (Modolell, Simpson, Klein) who worked on a slightly different context (proneural clusters of the dorsal thorax). It could be appropriate to revisit this view also in this context (or at least justify why conclusions obtained in the eye necessarily applies to another developmental context).

As suggested we have examined the proneural clusters of the dorsal thorax and explain how our Findings might apply there (subsection “Ato is required for altering Da and Emc levels in the morphogenetic furrow”, Figure 5BD), as well as to the proneural region of the anterior wing margin (subsection “Ato is required for altering Da and Emc levels in the morphogenetic furrow” and Figure 4I-H). We think our Findings apply in all three developmental contexts, but macrochaetae differ in that Emc and Da levels are modulated only in SOP cells, not in the proneural groups, at least at the stages we have examined. Troost et al. have described lower Emc in notum proneural regions but we describe this differently (subsection “AS-C is required for altering Da and Emc levels in wing disc proneural regions”, Figure 5B-D). Unlike the morphogenetic furrow or wing margin, where Emc protein virtually disappears, Emc levels in notum proneural regions seem the same as those in non-neural regions of the wing disc or eye disc. When Troost et al. describe them as ‘lower’ they appear to mean lower than certain other regions of the notum where Emc levels are in fact unusually high. Also, the ‘lower’ Emc (ie normal Emc) regions of the notum do not correspond to the proneural regions but are larger: we think they correspond to the general ground state of the notum which includes proneural regions as well as nonproneural regions.

Regarding the prepattern view of Emc proposed by others (Modolell, Simpson, Klein), what is the evidence for it exactly? It seems to be that Emc loss of function mutants have a phenotype and that Emc expression has a pattern. In our view the new Findings regarding ubiquitous Emc expression require re-evaluation of these views.

Reviewer #3:Li and Baker present a thorough analysis of the interdependence of the expression levels of two interacting HLH proteins, Da (a DNA-binding activator) and Emc (a non-DNA-binding inhibitor of Da). Earlier work from the same group had shown that Da activates emc expression, which then feeds back to repress da transcription by preventing Da from activating a da autoregulatory enhancer (Bhattacharya et al., 2011). The complex Emc expression pattern was thought to provide a prepattern for neural differentiation, in the sense that Da/proneural activators would be more active in regions of lowest emc expression. The authors claim that emc does not set the prepattern for neurogenesis, but instead its complex expression pattern depends to a great extent on the activity of Da and its proneural patterns.The crucial findings of Li and Baker are:1) Da protein levels are proportional to the da gene dosage in the range of 0-2. However, Emc protein levels do not similarly reflect the emc gene dosage (range of 0-2), instead they also reflect the da gene dosage! Da levels are not affected by emc gene dosage in the range of 1-2, but are greatly upregulated in emc null clones, due to the above-mentioned release of inhibition.2) The unexpected dependence of Emc protein levels on Da protein levels probably comes from the dramatic stabilization of Emc by Da, as determined by half-life measurements.3) The upregulation of Da and concomitant downregulation of Emc seen in proneural regions of the eye and wing margin need the expression of proneural bHLH proteins with intact HLH domains, but not necessarily DNA binding activity. This suggests that a proneural partner displaces Emc from Da, which leads to degradation of Emc and auto-activation of Da. Some unknown factor must also contribute to this process, since ectopic expression of proneural genes in non-proneural regions cannot recapitulate this effect on Emc and Da levels.4) emc transcription is not affected by ato or da mutations (exception: there is a moderate effect of da mutations only in the posterior part of the eye disk).This three-way post-transcriptional interplay among Da, Emc and proneurals represents an important in vivo demonstration of properties of HLH proteins that had only been shown in vitro. It also paves the way for deciphering the biological consequences of these important HLH heterodimerization properties, adding the interesting twist of stability regulation. That said, I have a few concerns that I will list below:a) The authors focus their "narrative" on how Emc does not represent a proneural prepattern, but rather reflects a proneural "response". I would say that it is both. They themselves show that the complex RNA pattern of emc does not depend (much) on Da or proneural factors, so this is patterned by something else. Granted, this complex pattern is greatly "evened out" at the protein level, thanks to the mutual interactions described by the authors. Still some proneural cluster "lows" in Emc protein (e.g. the morphogenetic furrow) have a clear proneural-independent transcriptional component, so they are expected to predispose these areas to become neurogenic, namely act as a prepattern. I would change the narrative in such a way to clarify that emc contributes to the prepattern (it cannot be the entire prepattern, since there is still some patterning of bristles in the absence of emc), but at the same time it is itself modulated by proneural activity (at the protein level). After all, whether Emc is or is not a prepatterning factor seems of lesser importance than the dissection of its molecular mode of action.

We do agree that Emc may still contribute to the prepattern mechanism and discuss this in the Discussion section. Surely, however, the origin of the spatial pattern of the prepattern is the key question? Our Findings undermine the idea that the Emc expression pattern is the origin of the prepattern.

b) They should point out that the previously reported activation of emc transcription by da was a misinterpretation of the destabilization of Emc in da null clones and that instead Da only affects transcription of emc behind the morphogenetic furrow (and that only modestly, as shown in Figure 7—figure supplement 1).

Since we have removed direct experiments on emc transcription from the paper (they will be submitted elsewhere), it is now less important to comment on the previous transcriptional model, although we do so briefly (subsection “Emc is stabilized by Da in S2 cells”).

c) They should point out that the dependence of Emc on Da levels does break down after a certain threshold. Overexpression of emc in Figure 2O-R causes an enormous increase of Emc protein without concomitantly increasing Da. Probably the degradation machinery of Emc is saturated. Intriguingly Da may also play some role even at these unnaturally high Emc levels, since in the furrow, where Da is engaged by Ato, this excess Emc seems to accumulate less. Therefore, its high accumulation in non-proneural regions is not solely because of saturation of the degradation machinery, since the latter seems to work better in the furrow.

We agree that the dependence of Emc on Da breaks down above a certain threshold and discuss this in the Discussion section.

d) The language needs to be improved. There are many places where the text is hard to follow and the meaning difficult to extract. Probably, with a thorough rewriting, the scientific points (a-c) above will also become clearer.

We agree that the previous manuscript was not as clear as desirable and believe

the revised manuscript is improved significantly.

[Editors' note: the author responses to the re-review follow.]

Reviewer #1:The revised manuscript is much improved, especially as it is more focused on the main message. The new experiment showing that uniform moderate levels of EMC driven by an exogenous promoter rescue neurogenesis certainly support the conclusion that post-transcriptional modulation of EMC is key to patterning neurogenesis. The new data showing that Da protein levels increase linearly with gene dosage are more convincing. Finally, the in vivo data showing that availability of Da regulates EMC protein levels are very nice.What remains rather circumstantially substantiated in my view is their mechanistic model of the regulation of EMC protein levels by Proneural proteins by competition for Da. I agree with the authors that the few data they have in this regard do not contradict this notion, and I in fact like the idea mostly because it fits the strong response of EMC to Da levels within the proneural domain under wild type conditions. I am surprised that they were unable to express Atonal in S2 cells to test their model more directly because expression of Ato in vitro has been reported previously.

A citation to the prior expression of Atonal in *Drosophila* cells would be most helpful, as we have yet to find such a report

The idea that the proneural pre-pattern depends on Da and Proneural proteins, not ID proteins is novel and interesting. Furthermore, it fits nicely with the emerging paradigm that post-transcriptional/translational modification of bHLH proteins is a key determinant of neurogenesis. The authors could do a better job referencing that literature in their Discussion section.

This is a good point. We added a sentence about post-translational modification of proneural proteins and a reference to an upcoming review (Discussion section).

In summary, the authors' model is the simplest and most parsimonious explanation of their data, but their advance of a specific mechanism for their observations remains relatively weak.Reviewer #2:This paper challenges the commonly held view that the ID factor Emc acts as a prepattern regulator of proneural activity during neurogenesis in Drosophila. Emc is known to heterodimerize with the E protein Da, thereby blocking its DNA-binding activity, hence its ability to regulate proneural gene expression. This paper reports on an alternative model whereby Da would stabilize Emc in the absence of proneural factor and proneural factors, patterned independently of Emc activity, would negatively regulate this stabilizing activity of Da towards Emc to generate the observed pattern of Emc protein accumulation.Several interesting observations support this model. First, in contexts of low proneural activity (S2 cells, wing pouch), the accumulation levels of Emc proteins appear to depend on Da levels. The analysis of Emc and Da levels in da, emc clones is really nice (Figure 2) and the stabilization of Emc by Da in S2 cells is clear (Figure 3).Second, in the proneural region of the eye, Ato is required for the accumulation of Da and the down-regulation of Emc in neural cells (Figure 4). The analysis of the ato1 and ato3 mutations is convincing. The observation that Emc is stabilized in the absence of Ato (Figure 4D and Figure 6) is interesting. However, how Ato destabilizes Emc is not understood: Ato does not appear to be sufficient to destabilize Emc (Figure 7). It is also not clear whether Ato acts indirectly via Da.Third, non-patterned expression of the emc gene largely rescues the emc mutant phenotype (Figure 8), indicating that Emc transcription does not provide key patterning cues.In summary, the strength of this paper is that it convincingly shows that the current model needs to be revised. Further experiments are, however, needed to strengthen the alternative model proposed here.Essential revisions:1) It would be nice to confirm the changes seen in Da levels by IF using Western blots, for instance by comparing Da levels in emc+/+ vs emc+/- discs (and Emc in da+/+ vs da+/- tissues). Please test.

We attempted to measure Da levels by western blot using a monoclonal antibody and to measure Emc levels using anti-GFP labeling of the Emc-YFP strain. Unfortunately, we were unable to detect endogenous levels of either protein by western blotting and so could not perform these experiments.

2) Figure 2E: it is not clear whether the increased Da levels seen in emc mutant cells result from protein stabilization (indicative of Emc targeting Da for degradation, as suggested by Figure 3F) or from increased transcription (due for instance to ectopic/premature expression of proneural factors). Please clarify.

We also think this is an interesting question and have been trying to assess experimentally whether Da is primarily restrained at the transcriptional level, as suggested previously (Bhattacharya and Baker, 2011), or at the level of protein stability, as Figure 3F suggests might be possible. So far, we do not have a uniform conclusion. We added a sentence to the paper raising the possibility that Da stability might be relevant to levels in vivo (subsection “Emc is stabilized by Da in S2 cells”).

3) Figure 3G: please control that nub>da does not affect emc gene expression (in situ, enhancer-trap…)

We have shown previously that ectopic Da expression in fact does elevate *emc* transcription (Bhattacharya et al., 2011, Figure 2C). Another reviewer previously asked us now to show that ectopic Da also leads to more Emc protein, consistent with our current model, and that is why we added Figure 3G to the manuscript. We could also remove Figure 3G again if preferred, but since it was requested by a previous reviewer we added to the figure legend that higher *emc* transcription is expected in this experiment.

4) Subsection “Ato is required for altering Da and Emc levels in the morphogenetic furrow”. The conclusion that Da up-regulation is transcriptional and bHLH-mediated needs to be supported by experimental evidence. Could the authors exclude the possibility that Emc also targets Da for degradation (see Figure 3F)?

The experimental data do not lead to a uniform conclusion at present (see point 2 above). Therefore, we changed the text in subsection “Ato is required for altering Da and Emc levels in the morphogenetic furrow” to include both possibilities.

5) Subsection “AS-C is required for altering Da and Emc levels in wing disc proneural regions”: This conclusion is not convincing. I find it hard to compare Da levels based on different IF experiments. Clonal analysis is required.

We agree with the reviewer, and this is why we based our definitive conclusions on clonal analysis with a complete deletion of AS-C in Figures 4H,I. We only included sc[10-1] because a reviewer of the previous manuscript asked that we explore the conclusions published by other groups regarding notum patterning in light of our findings. Since those other studies have used the viable sc[10-1] allele, we also included it here for comparison. We could not find any sc[10-1] FRT strain available, and although one should be easy to generate we could not have done so and performed the clonal analysis within two months. We could remove this sc[10-1] data, but in our opinion it might as well be included. We modified the Results section to indicate that a clonal analysis with sc[10-1] could provide stronger results (subsection “AS-C is required for altering Da and Emc levels in wing disc proneural regions”).

6) Subsection “AS-C is required for altering Da and Emc levels in wing disc proneural regions”: Whether AS-C regulates Emc/Da levels in sensory cells is difficult to study in the absence of these cells. The data shown in Figure 5B-D are not convincing.

The cells are present, but not specified as sensory cells. As such there are no markers that could identify them for double labeling, so this cannot be further clarified experimentally. We agree that the failure to find single cells with lower Emc could be equivocal, but we do think it is more significant that we cannot find the single cells with higher Da. Figure 5D shows that we can find these high Da/low Emc SOP cells in the wild type. We changed the text of the Results and Discussion sections to make the uncertainties clear (subsection “AS-C is required for altering Da and Emc levels in wing disc proneural”, Discussion section). This is another example where we only present data on the notum in response to the previous reviewer that requested this.

7) Figure 7B: heterogenous levels of Ato appear to be detected in the non-proneural region (Figure 7B). Therefore, a per-cell quantification of the relative Emc/Da levels is needed (Figure 7A). It would be nice to test whether Ato levels correlate with Emc/Da levels. Also, what is causing the observed heterogeneity in Ato levels?

We performed a per-cell analysis as requested. The data are shown in Figure 7 panels C and D and Figure 7—figure supplement 1C and 1D and described in subsection “Proneural genes are not sufficient to regulate Da or Emc protein levels.”). We saw no evidence that Emc levels were lower in the scattered cells with higher Ato levels. We did see a weak trend towards more Da in high Ato-cells, which is consistent with the slightly higher Da levels already noted in cells expressing ectopic Ato (Figure 7A and Figure 7—figure supplement 1A).

The cause of the heterogenous Ato levels is a mystery at present.

8) Figure 7—figure supplement 1A: the clone of act>ato cells (right) exhibits high Emc and high Da levels. This seems to contradict the model whereby Ato competes Emc for binding Da. Please comment.

Yes, this is part of the data for suggesting that Ato is not sufficient to degrade Emc at ectopic locations (subsection “Proneural genes are not sufficient to regulate Da or Emc protein levels.”). Since ectopic Ato expression in wing discs significantly alters wing disc development, perhaps some effects on Emc and Da levels could be indirect. Overall, however, the data do not clearly support the notion that Ato leads to Emc degradation at ectopic locations.

9) Figure 8: what is the phenotype of emcD1/emcAP6 flies? are these mRNA null? what is the emc mRNA pattern in the mutant and rescued eye discs? Is the latter similar to the protein pattern shown in 8C? does the UAS-ems construct contain the 3'UTR of the emc gene? Does it matter? Also, is the emc mutant combination used in Figure 8 protein null?

We added brief description of the emc^∆1^ allele and the emc^AP6^/emc^∆1^ genotype (subsection “Uniform Emc transcription supports neural patterning”; subsection “*Drosophila* Strains”). emc^∆1^ is an early frame-shift allele and protein null, one of two such alleles we generated by Crispr. We will describe these alleles in detail elsewhere, we do not believe it is of sufficient interest to do so here. We used the transheterozygous emc^AP6^/emc^∆1^ genotype for the rescue experiments to exclude the possibility of linked mutations on the extant *emc^AP6^* chromosome. We did not sequence the UAS-emc5.3 construct. This was described by Baonza et al., (2000) as “a full-length *emc* cDNA” so it might include endogenous 3’ UTR that could encode post-transcriptional regulation eg by miRNAs. Therefore we are careful to refer only to uniform transcription in this experiment, although please note that the antibody stainings included in this paper show uniform protein expression in the relevant areas after uniform transcription.